# Simplifying and Stabilizing Model Selection in Unsupervised Domain Adaptation

## Abstract

Unsupervised domain adaptation (UDA) is a potent approach for enhancing model performance in an unlabeled target domain by leveraging relevant labeled data from a source domain. Despite the significant progress in UDA facilitated by deep learning, model selection, already a challenging task with deep models, becomes considerably more demanding in UDA scenarios due to the absence of labeled target data and substantial distribution shifts between domains. Existing model selection methods in UDA often struggle to maintain stable selections across diverse UDA methods and various UDA scenarios, frequently resulting in suboptimal or even the worst choices. This limitation significantly impairs their practicality and reliability for researchers and practitioners in the community. To address this challenge, we introduce a novel ensemble-based validation approach called EnsV, aiming to simplify and stabilize model selection in UDA. EnsV relies solely on predictions of unlabeled target data without making any assumptions about distribution shifts, offering high simplicity and versatility. Additionally, EnsV is built upon an off-the-shelf ensemble that is theoretically guaranteed to outperform the worst candidate model, ensuring high stability. In our experiments, we compare EnsV to 8 competitive model selection approaches. Our evaluation involves 12 UDA methods across 5 diverse UDA benchmarks and 5 popular UDA scenarios. The results consistently demonstrate that EnsV stands out as a highly simple, versatile, and stable approach for practical model selection in UDA scenarios.

## 1 Introduction

Deep learning models (Krizhevsky et al., 2012; He et al., 2016; Dosovitskiy et al., 2021) have achieved significant advancements in various tasks through supervised learning with large labeled datasets (Russakovsky et al., 2015). However, obtaining labels can be expensive, and deep models often struggle to generalize to unlabeled data from unseen distributions (Hendrycks & Gimpel, 2016). Domain adaptation (Pan & Yang, 2009) offers a promising solution to this challenge by transferring knowledge from a labeled source domain to a target domain with limited labels but a similar task. In particular, unsupervised domain adaptation (Pan et al., 2010) (UDA) has gained significant attention due to its practical assumption that the target domain is entirely unlabeled.

Initially, UDA is studied in a closed-set setting (CDA) where only covariate shift (Sugiyama et al., 2007) is considered as the domain shift, and the two domains share the same label set. Recent research has explored many real-world UDA scenarios by incorporating label shift, where the two domains have distinct label sets. This includes partial-set UDA (PDA) (Cao et al., 2018), where several source classes are missing in the target domain, open-set UDA (ODA) (Panareda Busto & Gall, 2017), where the target domain contains samples from unknown classes, and open-partial-set UDA (OPDA) (Saito et al., 2020), where there are only some overlaps in the label sets across domains. More recently, source-free UDA settings (SFUDA) (Li et al., 2020; Liang et al., 2020) have been explored, where only the source model instead of source data is available for target adaptation, potentially addressing privacy concerns in the source domain. Subsequently, in the context of black-box domain adaptation (Liang et al., 2021), the privacy of the source domain is fully safeguarded.

Specifically, the research community has made significant efforts to develop effective UDA methods in image classification (Ganin & Lempitsky, 2015; Long et al., 2018) and semantic segmentation (Tsai et al., 2018; Vu et al., 2019), which can be seen through two distinct research directions. The first

direction focuses on aligning the distributions across domains by minimizing specific discrepancy measures (Gong et al., 2012; Fernando et al., 2013; Long et al., 2015; Sun & Saenko, 2016; Yang & Soatto, 2020) or using adversarial learning to maximize domain confusion (Ganin & Lempitsky, 2015). Especially, adversarial learning has become a popular approach and has been explored at different levels for domain alignment, including image-level (Hoffman et al., 2018), manifold-level (Ganin & Lempitsky, 2015; Tzeng et al., 2017; Long et al., 2018), and prediction-level (Saito et al., 2018; Tsai et al., 2018; Vu et al., 2019; Zhang et al., 2019). The second direction focuses on target-oriented learning, aiming to learn a suitable structure for the target domain. This includes self-training approaches (Shu et al., 2018; Liang et al., 2020; 2021) and target-specific regularizations (Xu et al., 2019; Cui et al., 2020; Jin et al., 2020).

While UDA has witnessed significant advancements, the successful application of UDA methods across diverse tasks relies heavily on selecting appropriate hyperparameters. Sub-optimal hyperparameters can cause state-of-the-art UDA methods to underperform compared to the source model without adaptation (Saito et al., 2021; Musgrave et al., 2022), emphasizing the significance of model selection or hyperparameter selection in UDA. In a typical model selection scenario, we are presented with a set of $m$ candidate models with the weights $\{\theta_i\}_{i=1}^m$. These models are trained using a given UDA method with a corresponding set of hyperparameters $\{\eta_i\}_{i=1}^m$. The goal is to identify the candidate model that exhibits the best performance on the unlabeled target domain and subsequently adopt the associated hyperparameters. This model selection problem remains challenging and under-explored in UDA due to significant domain shifts and the absence of labeled target data. Existing approaches can be categorized into two types. The first type involves leveraging labeled source data for target model selection (Sugiyama et al., 2007; You et al., 2019; Ganin et al., 2016). The second type designs unsupervised metrics based on priors of the learned target structure and utilizes them for selection (Morerio et al., 2017; Saito et al., 2021; Musgrave et al., 2022; Tu et al., 2023). Despite their particular designs, all of these methods face challenges in avoiding the selection of poor models or even the worst models across various UDA methods and settings, limiting their adoption by researchers and practitioners in the community (Musgrave et al., 2022).

In this paper, we aim to address this dilemma by introducing a novel ensemble-based validation approach, called EnsV. Our approach originates from a meticulous examination of the model selection problem itself. Surprisingly, we discovered that the problem setting inherently provides an ensemble of candidate models without any additional effort. Unfortunately, many existing model selection studies overlook this "free lunch", treating each candidate model independently. Our theoretical analysis of the ensemble confirms that it consistently outperforms the worst candidate model. Motivated by the ensemble's strength, we propose EnsV, which leverages the ensemble as a role model for the direct assessment of candidate models. EnsV tackles model selection exclusively with target predictions generated by all candidate models, eliminating the need for specific domain shift assumptions and source data access. This simplicity and versatility make it suitable for various UDA tasks. The role model utilized in EnsV, backed by performance guarantees, reinforces the robust model selection stability that EnsV offers. **Our main contributions** can be summarized as follows:

- We study the significant but under-explored problem of model selection in UDA. To the best of our knowledge, we are the first to approach it through the lens of ensembles. Furthermore, we uncover an intriguing "free lunch" of ensembles in model selection, substantiated by theoretical and empirical evidence of their superiority over the worst candidate model.

- With a novel perspective on approximating target ground truth labels for model selection, we introduce a novel ensemble-based validation approach, known as EnsV. In EnsV, we ingeniously leverage the performance-guaranteed ensemble as a reliable role model. This allows us to effortlessly select the best candidate model through a straightforward accuracy comparison. Notably, both steps within EnsV rely solely on the predictions of unlabeled target data, requiring no additional complexities.

- We conduct a comprehensive empirical study to compare EnsV's model selection performance with that of existing methods. This study encompasses the evaluation of 8 model selection approaches, 12 UDA methods, 5 UDA benchmarks, and 5 practical UDA scenarios. The results consistently showcase EnsV's superiority, as it consistently achieves the best model selection performance on average while effectively avoiding the selection of poor models. This solidifies its position as a simple, versatile, and highly stable baseline for model selection in various UDA scenarios.

Table 1: Comparing existing methods for model selection in unsupervised domain adaptation.

| Validation Method | Covariate Shift | Label Shift | w/o source data | w/o hyperparameters | w/o extra training |
|---|---|---|---|---|---|
| SourceRisk | ✗ | ✗ | ✗ | ✗ | ✓ |
| IWCV | ✓ | ✗ | ✗ | ✗ | ✗ |
| DEV | ✓ | ✗ | ✗ | ✗ | ✗ |
| RV | ✓ | ✗ | ✗ | ✗ | ✗ |
| Entropy | ✓ | ✗ | ✓ | ✓ | ✓ |
| InfoMax | ✓ | ✗ | ✓ | ✓ | ✓ |
| SND | ✓ | ✓ | ✓ | ✗ | ✓ |
| Corr-C | ✓ | ✗ | ✓ | ✓ | ✓ |
| EnsV (Ours) | ✓ | ✓ | ✓ | ✓ | ✓ |

## 2 RELATED WORK

**Model selection** in unsupervised domain adaptation (UDA) is significant in the practical deployment of UDA methods but remains relatively under-explored. Efforts to address this challenge can be broadly categorized into two lines. Early approaches to model selection in UDA focused on estimating the target domain risk through labeled source data. SourceRisk Ganin & Lempitsky (2015) utilized a hold-out source validation set to guide model selection based on source risk. To mitigate the impact of domain shift on source estimation, Sugiyama et al. (2007) introduced Importance-Weighted Cross Validation (IWCV), which re-weights source risk using a source-target density ratio estimated in the input space. Building upon this, You et al. (2019) improved IWCV by introducing Deep Embedded Validation (DEV), which estimates the density ratio in the feature space and offers lower variance. Ganin et al. (2016) proposed a novel Reverse Validation approach (RV) that leveraged reversed source risk for selection. However, source-based validation methods often necessitate additional model training to handle domain shifts, rendering them cumbersome and less reliable. In contrast, recent model selection methods have shifted their focus exclusively to unlabeled target data, employing specifically designed metrics for model selection. For instance, Morerio et al. (2017) introduced the mean Shannon's Entropy of target predictions as a model selection metric, promoting confident predictions. Musgrave et al. (2022) proposed the use of Input-Output Mutual Information Maximization (InfoMax)Bridle et al. (1991) as a metric, augmented with class-balance regularization over Entropy. Saito et al. (2021) introduced Soft Neighborhood Density (SND), a novel metric focusing on neighborhood consistency. Tu et al. (2023) presented Corr-C, a class correlation-based metric that evaluates both class diversity and prediction certainty simultaneously. Our EnsV approach aligns with the latter line of research. EnsV approaches the model selection problem from a novel perspective, leveraging the power of the inherent ensemble. Importantly, it operates without making any assumptions about distribution shifts or the learned target structure, making it suitable for a variety of UDA scenarios. A comprehensive comparison, as presented in Table 1, underscores that EnsV stands out as a simple and versatile approach.

**Ensemble** methods, which harness the collective power of a pool of models through prediction averaging, have been extensively studied in the machine learning community for enhancing model performance (Perrone & Cooper, 1995; Opitz & Maclin, 1999; Bauer & Kohavi, 1999; Dietterich, 2000) and improving model calibration (Lakshminarayanan et al., 2017; Ovadia et al., 2019). In the era of deep learning, the efficiency of ensembling has garnered significant attention due to the high training cost of deep models. Efficient solutions have been proposed, such as using partially shared parameters (Lee et al., 2015; Wen et al., 2020; Dusenberry et al., 2020) and leveraging intermediate snapshots (Huang et al., 2017; Garipov et al., 2018; Benton et al., 2021). Recently, weight averaging has gained attention as an efficient alternative to prediction averaging during inference (Izmailov et al., 2018; Wortsman et al., 2022; Matena & Raffel, 2022; Rame et al., 2022; Ramé et al., 2022). In addition, diversity is considered crucial for effective ensembles. Various approaches have been explored to achieve diverse checkpoints, including bootstrapping (Freund et al., 1996), random initializations (Fort et al., 2019), tuning hyperparameters (Wenzel et al., 2020; Zaidi et al., 2021; Wortsman et al., 2022), and combining multiple strategies (Gontijo-Lopes et al., 2021). Different from existing ensemble applications, our work innovatively and elegantly applies ensemble to help address the open problem of unsupervised model selection in domain adaptation.

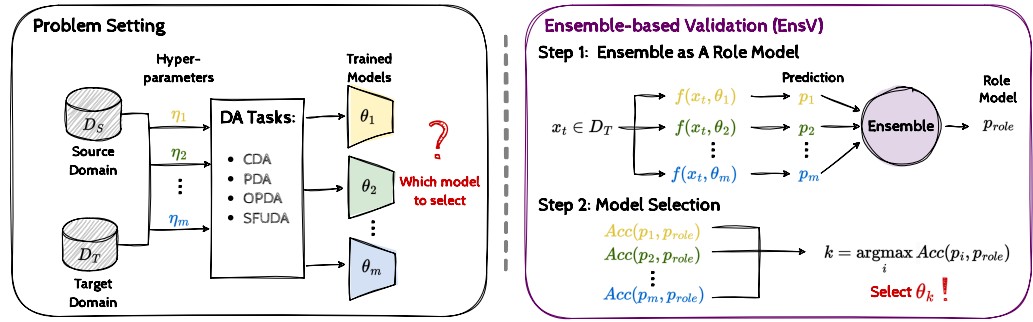

Figure 1: Overview of our model selection approach EnsV for unsupervised domain adaptation.

## 3 METHODOLOGY

We consider a $C$-way image classification task to introduce the concept of unsupervised domain adaptation (UDA). In UDA, we typically have a labeled source domain $\mathcal{D}_\mathrm{s} = \{(x_\mathrm{s}^i, y_\mathrm{s}^i)\}_{i=1}^{n_\mathrm{s}}$ comprising $n_\mathrm{s}$ annotated source images $x_\mathrm{s}$ and their corresponding labels $y_\mathrm{s}$. Additionally, there is an unlabeled target domain, $\mathcal{D}_\mathrm{t} = \{x_\mathrm{t}^i\}_{i=1}^{n_\mathrm{t}}$, containing only $n_\mathrm{t}$ unlabeled target images $x_\mathrm{t}$. Despite the tasks being similar, there exist data distribution shifts between the two domains. The primary objective of UDA is to accurately predict the unavailable target labels, $\{y_\mathrm{t}^i\}_{i=1}^{n_\mathrm{t}}$, by leveraging a discriminative mapping $f(x, \theta)$, which is learned using data from two domains. Here, $\theta \in \mathbb{R}^d$ represents the weights of the trained UDA model. When presented with an input image $x$, the model generates a probability prediction vector, $p = f(x, \theta)$, where $p \in \mathbb{R}^C$ and $\sum_{i=1}^C p^i = 1$.

The model selection problem in UDA is essentially equivalent to the hyperparameter selection challenge. Here, we aim to determine the optimal hyperparameter $\eta$ from a set of $m$ candidate values $\{\eta_i\}_{i=1}^m$. The hyperparameter $\eta$ can encompass various aspects, including the learning rate, loss coefficients, architectural settings, training iterations, and more. By training UDA models using the $m$ different values of $\eta$, we obtain corresponding models with weights denoted as $\{\theta_i\}_{i=1}^m$. In UDA, the objective of model selection is to pinpoint the model $\theta_k$ that demonstrates the best performance on the unlabeled target domain. Subsequently, we select the corresponding hyperparameter $\eta_k$ as the optimal choice for potential adaptation with unlabeled target samples from the exact target domain. We illustrate the problem setting in Figure 1. Without loss of generality, in this paper, we assume $m$ is greater than 1, and candidate models have different weights $\theta$, resulting in different discriminative mappings of $f(x, \theta)$. For clarity, we treat both $\theta$ and the model interchangeably in our presentation. This also applies to model selection, hyperparameter selection, and validation.

### 3.1 ENSEMBLE: THE OVERLOOKED "FREE LUNCH" IN MODEL SELECTION

Model selection in UDA is challenging due to the absence of labeled target data available for directly evaluating candidate models. Existing approaches typically address this challenge from two perspectives: leveraging the help of labeled source data (You et al., 2019) and utilizing specific prior assumptions to design unsupervised metrics (Saito et al., 2021). Surprisingly, we've observed that all existing model selection methods treat each model independently, overlooking the collective potential offered by the off-the-shelf ensemble created by these candidates. In this paper, unless otherwise specified, the ensemble refers to prediction-based ensembling, which involves averaging probability predictions across all models to obtain the averaged prediction, i.e., $\frac{1}{m} \sum_{i=1}^m f(x, \theta_i)$ for a sample $x$.

Contrastingly, we adopt a fresh perspective in analyzing the challenge of model selection in UDA by leveraging the ensemble. Typically, two concerns arise when considering the use of the ensemble: one pertains to the efficiency issue caused by training multiple models, and the other relates to the lack of diversity among candidate models. In the context of model selection, we observe that, without the need to introduce additional models, the problem setting inherently provides a range of pre-existing candidate models, directly addressing the efficiency concern. Furthermore, all candidate models are trained using a UDA method with varying hyperparameters, yielding diverse yet effective discriminative abilities. This naturally eases the diversity concern. As a surprising consequence, the ensemble appears to be a "free lunch" in the context of UDA model selection, a point that has been

previously overlooked by researchers. To gain a deeper insight into the effectiveness of the ensemble, we present a theoretical analysis grounded in the proposition below.

**Proposition 1** *Given the use of negative log-likelihood (NLL) as the loss function, defined as $l(p, y) = -\log p^y$, and considering a random target sample $x$ with label $y$, the following inequality can be established between the loss of the ensemble $\frac{1}{m} \sum_{i=1}^{m} f(x, \theta_i)$, the averaged loss of all candidate models $\{\theta_i\}_{i=1}^{m}$, and the loss of the worst candidate model $\theta_{\text{worst}}$:*

$$l(\frac{1}{m} \sum_{i=1}^{m} f(x, \theta_i), y) < \frac{1}{m} \sum_{i=1}^{m} l(f(x, \theta_i), y) < l(f(x, \theta_{\text{worst}}), y).$$

Kindly refer to Appendix A for proof of the above proposition. This proposition theoretically guarantees that the ensemble always outperforms the worst candidate model. In contrast, as demonstrated by our comprehensive experiments, existing model selection methods cannot consistently avoid selecting the worst candidate model.

### 3.2 ENSEMBLE AS A ROLE MODEL: SIMPLE AND STABLE MODEL SELECTION

When it comes to addressing model selection in UDA, an oracle solution would involve selecting models based on their accuracy, measured against the unattainable target ground truth $\{y_t^i\}_{i=1}^{n_t}$. Motivated by this ideal case, we offer a novel perspective on model selection: If we can obtain a reliable approximation of the true target labels, we can use it directly for accurate model selection. To achieve this, we leverage the off-the-shelf yet performance-guaranteed ensemble and take a further step by using it to build a reliable role model. We then select the model that most closely resembles this role model among all candidates. With these two straightforward steps, we introduce an elegantly simple model selection approach known as ensemble-based validation (EnsV).

**Step 1: Ensemble as a role model.** To begin with, for each unlabeled target sample $x$, we consider the ensemble $\frac{1}{m} \sum_{i=1}^{m} f(x, \theta_i)$ as a reliable estimation of its ground truth. This enables us to obtain reliable predictions for all target data, denoted as $\{\frac{1}{m} \sum_{i=1}^{m} f(x_j, \theta_i)\}_{j=1}^{n_t}$. These ensembles serve as our role model, providing guidance for accurate model selection in the subsequent step.

**Step 2: Model selection.** In this step, we utilize the role model to assess all candidate models and select the one with the highest similarity. For simplicity, EnsV involves a direct measurement of accuracy between the role model $\{\frac{1}{m} \sum_{i=1}^{m} f(x_j, \theta_i)\}_{j=1}^{n_t}$ and the predictions made by each candidate model, such as $\{f(x_j, \theta_i)\}_{j=1}^{n_t}$ for the model with weights $\theta_i$. We then select the model $\theta_k$ with the highest accuracy and determine the optimal value $\eta_k$ for the hyperparameter $\eta$.

We present a comprehensive illustration of our ensemble-based validation approach, known as EnsV, in Figure 1. Furthermore, through a comparison with other model selection methods presented in Table 1, we can observe that EnsV stands out as a simple but versatile validation method.

## 4 EXPERIMENTS

### 4.1 SETUP

**Datasets.** Our experiments encompass diverse and widely-used image classification benchmarks: (*i*) *Office-31*(Saenko et al., 2010) with 31 classes and 3 domains (Amazon (A), DSLR (D), and Webcam (W)); (*ii*) *Office-Home*(Venkateswara et al., 2017) with 65 classes and 4 domains (Art (Ar), Clipart (Cl), Product (Pr), and Real-World (Re)); (*iii*) *VisDA*(Peng et al., 2017) with 12 classes and 2 domains (training (T) and validation (V)); and (*iv*) *DomainNet-126*(Peng et al., 2019; Saito et al., 2018) with 126 classes and 4 domains (Real (R), Clipart (C), Painting (P), and Sketch (S)). Additionally, we conduct experiments in synthetic-to-real semantic segmentation, specifically targeting the transfer from GTAV(Richter et al., 2016) to Cityscapes(Cordts et al., 2016).

**UDA methods.** In our experiments, we assess all the model selection methods listed in Table 1. Kindly see Appendix C for more introductions. With these validation methods, we perform model selection for various UDA methods across different UDA settings. For CDA, we consider ATDOC (Liang et al., 2021), BNM (Cui et al., 2020), CDAN (Long et al., 2018), MCC (Jin et al., 2020), MDD (Zhang

Table 2: CDA accuracy (%) on *Office-Home* (*Home*). **bold**: Best value.

| Method | ATDOC | | | | | BNM | | | | | CDAN | | | | |
|---|---|---|---|---|---|---|---|---|---|---|---|---|---|---|---|
| | →Ar | →Cl | →Pr | →Re | avg | →Ar | →Cl | →Pr | →Re | avg | →Ar | →Cl | →Pr | →Re | avg |
| SourceRisk | 66.63 | 52.54 | 78.57 | 76.61 | 68.59 | 62.44 | 50.74 | 77.53 | 74.76 | 66.37 | 55.00 | 42.65 | 69.50 | 68.81 | 58.99 |
| IWCV | 67.97 | 54.03 | 78.31 | 79.26 | 69.89 | 66.56 | 48.16 | 74.09 | 73.28 | 65.52 | 61.31 | 41.24 | 67.17 | 71.93 | 60.41 |
| DEV | 67.39 | 54.23 | 77.78 | 79.39 | 69.70 | 65.76 | 56.39 | 73.92 | 77.59 | 68.41 | 67.23 | 57.04 | 68.76 | 76.91 | 67.49 |
| RV | 68.68 | 56.13 | 78.93 | 79.64 | 70.85 | 68.25 | 56.75 | **78.08** | 78.67 | 70.44 | 67.66 | 56.74 | 76.01 | 77.68 | 69.52 |
| Entropy | 63.67 | 55.83 | 76.54 | 78.36 | 68.60 | 66.28 | 54.49 | 74.15 | 77.64 | 68.14 | 67.66 | **57.56** | 76.37 | 77.45 | 69.76 |
| InfoMax | 63.67 | 55.63 | 77.61 | 78.36 | 68.82 | 66.28 | 54.49 | 74.15 | 77.64 | 68.14 | 67.66 | **57.56** | 76.37 | 77.45 | 69.76 |
| SND | 63.67 | 55.63 | 76.54 | 77.54 | 68.34 | 66.28 | 54.49 | 74.15 | 77.64 | 68.14 | **67.94** | **57.56** | 76.96 | 77.68 | 70.04 |
| Corr-C | 63.51 | 50.39 | 73.89 | 73.88 | 65.42 | 58.10 | 45.37 | 68.97 | 70.59 | 60.76 | 53.84 | 41.21 | 64.96 | 67.65 | 56.91 |
| EnsV | **68.70** | **58.05** | **79.81** | **80.41** | **71.74** | **68.61** | 57.38 | **78.08** | 79.54 | **70.90** | 67.88 | **57.56** | 77.39 | 78.19 | 70.25 |
| Worst | 62.89 | 50.39 | 73.89 | 73.88 | 65.26 | 58.10 | 45.37 | 68.96 | 70.59 | 60.75 | 53.80 | 41.21 | 64.78 | 67.65 | 56.86 |
| Best | 68.97 | 58.35 | 80.27 | 80.58 | 72.04 | 68.93 | 57.51 | 78.43 | 79.57 | 71.11 | 68.19 | 57.90 | 77.44 | 78.19 | 70.43 |

| Method | MCC | | | | | MDD | | | | | SAFN | | | | | *Home* |
|---|---|---|---|---|---|---|---|---|---|---|---|---|---|---|---|---|
| | →Ar | →Cl | →Pr | →Re | avg | →Ar | →Cl | →Pr | →Re | avg | →Ar | →Cl | →Pr | →Re | avg | AVG |
| SourceRisk | 66.57 | 56.53 | 79.55 | 80.90 | 70.89 | 62.53 | 54.43 | 75.27 | 75.55 | 66.94 | 63.54 | 51.34 | 73.66 | 74.54 | 65.77 | 66.26 |
| IWCV | 68.69 | 58.93 | 80.37 | 80.08 | 72.02 | 64.20 | 56.50 | 73.78 | 74.28 | 67.19 | 63.90 | **52.36** | 72.31 | 74.29 | 65.82 | 66.81 |
| DEV | 68.81 | 58.07 | 78.54 | 80.10 | 71.38 | 64.42 | 56.94 | 76.85 | 75.94 | 68.54 | 63.15 | 50.47 | 71.20 | 74.54 | 64.84 | 68.39 |
| RV | **70.40** | 58.80 | **80.63** | 80.39 | 72.56 | **66.57** | 55.75 | 76.60 | 76.90 | 68.96 | 64.31 | 50.13 | 73.77 | 74.93 | 65.78 | 69.68 |
| Entropy | 69.29 | 59.33 | **80.63** | 80.96 | 72.55 | 66.54 | 57.63 | 77.27 | **77.45** | 69.72 | 59.85 | 46.41 | 72.51 | 73.18 | 62.99 | 68.63 |
| InfoMax | 66.58 | 58.48 | 79.12 | 80.81 | 71.25 | 66.54 | 57.74 | 77.27 | **77.45** | **69.75** | 64.56 | 49.71 | 73.77 | 73.18 | 65.31 | 68.84 |
| SND | 69.05 | 55.61 | 79.72 | 79.10 | 70.87 | 51.34 | 38.01 | **77.61** | 68.46 | 58.86 | 57.90 | 46.41 | 67.04 | 68.18 | 59.88 | 66.02 |
| Corr-C | 69.05 | 55.61 | 79.72 | 79.10 | 70.87 | 47.79 | 31.69 | 63.40 | 60.63 | 50.88 | 62.66 | 46.41 | 68.83 | 68.18 | 61.52 | 61.06 |
| EnsV | 69.92 | **59.50** | 80.30 | 80.86 | **72.65** | 66.46 | **57.81** | **77.61** | 76.51 | 69.60 | **65.91** | 52.18 | **74.51** | 75.57 | **67.04** | **70.36** |
| Worst | 62.72 | 54.63 | 76.19 | 78.19 | 67.93 | 47.79 | 31.69 | 63.40 | 60.63 | 50.88 | 57.90 | 46.41 | 67.04 | 68.18 | 59.88 | 60.26 |
| Best | 70.68 | 59.95 | 80.93 | 81.02 | 73.14 | 66.75 | 58.36 | 77.61 | 77.45 | 70.04 | 66.59 | 53.14 | 74.90 | 75.57 | 67.55 | 70.72 |

Table 3: CDA accuracy (%) on *Office-31* (*Office*) and *VisDA*.

| Method | ATDOC | | | | | BNM | | | | | CDAN | | | | |
|---|---|---|---|---|---|---|---|---|---|---|---|---|---|---|---|
| | →A | →D | →W | avg | T→V | →A | →D | →W | avg | T→V | →A | →D | →W | avg | T→V |
| SourceRisk | 72.56 | 88.96 | **87.80** | 83.11 | 67.79 | 72.92 | **90.36** | **89.43** | 84.24 | 70.51 | 63.90 | 91.16 | **89.06** | 81.37 | 64.50 |
| IWCV | 72.56 | 86.14 | 86.54 | 81.75 | 67.79 | 72.92 | 85.54 | **89.43** | 82.63 | **76.94** | 63.90 | 91.16 | 88.30 | 81.12 | 64.50 |
| DEV | 72.56 | 86.14 | 86.54 | 81.75 | 70.34 | 72.92 | 85.54 | **89.43** | 82.63 | **76.94** | 63.90 | 91.16 | 88.30 | 81.12 | 64.50 |
| RV | **74.93** | 89.96 | 87.23 | 84.04 | **77.37** | 70.71 | 88.55 | **89.43** | 82.90 | 74.58 | **73.27** | 91.16 | 88.30 | 84.24 | 76.02 |
| Entropy | 73.29 | 86.14 | **87.80** | 82.41 | 62.85 | 72.67 | 85.54 | 83.14 | 80.45 | 58.36 | 71.62 | 91.16 | **89.06** | 83.95 | **80.46** |
| InfoMax | 73.29 | 86.14 | **87.80** | 82.41 | 76.49 | 70.52 | 85.54 | 83.14 | 79.73 | 58.36 | 71.62 | 91.16 | 88.30 | 83.69 | **80.46** |
| SND | 73.29 | **92.37** | **87.80** | 84.49 | **77.37** | 74.44 | 85.54 | 83.14 | 81.04 | 69.65 | 71.55 | 92.37 | 88.55 | 84.16 | **80.46** |
| Corr-C | 71.05 | 90.96 | 84.40 | 82.14 | 67.79 | 67.16 | 84.34 | 78.99 | 76.83 | 70.51 | 58.29 | 67.67 | 59.62 | 61.86 | 64.50 |
| EnsV | 74.83 | 90.96 | **87.80** | **84.53** | 73.36 | **74.87** | **90.36** | **89.43** | **84.89** | 74.58 | 73.20 | **92.77** | 88.55 | **84.84** | 79.05 |
| Worst | 71.05 | 86.14 | 84.40 | 80.53 | 62.85 | 67.16 | 84.34 | 78.99 | 76.83 | 23.08 | 58.29 | 67.67 | 57.11 | 61.02 | 64.50 |
| Best | 75.31 | 92.37 | 87.80 | 85.16 | 77.37 | 75.52 | 90.36 | 89.43 | 85.10 | 76.94 | 73.38 | 92.77 | 89.06 | 85.07 | 80.46 |

| Method | MCC | | | | | MDD | | | | | SAFN | | | | | *Office* | *VisDA* |
|---|---|---|---|---|---|---|---|---|---|---|---|---|---|---|---|---|---|
| | →A | →D | →W | avg | T→V | →A | →D | →W | avg | T→V | →A | →D | →W | avg | T→V | AVG | AVG |
| SourceRisk | 73.11 | 90.96 | 91.07 | 85.05 | 80.46 | 75.72 | 91.06 | 86.23 | 84.34 | 72.25 | 69.20 | 83.73 | 87.17 | 80.03 | 70.71 | 83.02 | 71.04 |
| IWCV | 73.11 | 91.16 | 88.55 | 84.27 | 81.48 | 75.49 | 91.16 | 89.18 | 85.28 | 72.25 | 69.32 | 86.55 | 80.38 | 78.75 | 66.33 | 79.43 | 71.55 |
| DEV | 72.70 | 89.16 | 93.08 | 84.98 | 81.48 | 75.65 | 91.16 | 89.18 | 85.33 | 72.25 | 68.69 | 90.83 | 87.17 | 82.23 | 66.33 | 82.36 | 71.97 |
| RV | **73.97** | 89.06 | 93.08 | 85.37 | **82.22** | 74.46 | **92.57** | 86.79 | 84.61 | 77.23 | 68.69 | 90.83 | 87.17 | 82.23 | 66.33 | 83.90 | 75.62 |
| Entropy | 73.93 | 90.56 | **93.46** | 85.98 | **82.22** | 76.31 | **92.57** | 90.82 | 86.57 | **78.95** | 68.23 | **91.57** | 85.66 | 81.82 | 70.20 | 83.53 | 72.17 |
| InfoMax | 73.93 | 89.16 | 88.55 | 83.88 | 81.48 | **76.50** | **92.57** | 90.82 | **86.63** | **78.95** | 68.23 | **91.57** | **87.42** | 82.41 | 70.20 | 83.13 | 74.32 |
| SND | 73.93 | **91.97** | **93.46** | **86.45** | 69.35 | **76.50** | 92.17 | 90.82 | 86.50 | **78.95** | 68.23 | 89.96 | 85.66 | 81.28 | 58.15 | 83.99 | 72.32 |
| Corr-C | 73.93 | 91.37 | **93.46** | 86.25 | 69.35 | 74.25 | 91.57 | 85.66 | 83.83 | 72.25 | 68.39 | 86.75 | 80.38 | 78.51 | 62.52 | 78.24 | 67.82 |
| EnsV | 73.75 | 90.56 | 91.45 | 85.25 | **82.22** | 75.92 | **92.57** | 90.82 | 86.44 | 77.23 | **69.67** | 90.96 | 87.17 | **82.60** | **73.96** | **84.76** | **76.73** |
| Worst | 70.56 | 86.75 | 87.17 | 81.49 | 69.35 | 73.06 | 87.35 | 85.66 | 82.02 | 72.25 | 67.27 | 83.73 | 80.38 | 77.13 | 58.15 | 76.50 | 58.36 |
| Best | 74.42 | 91.97 | 93.46 | 86.62 | 82.23 | 76.52 | 92.57 | 92.20 | 87.10 | 78.95 | 70.06 | 91.57 | 87.42 | 83.02 | 75.30 | 85.34 | 78.54 |

et al., 2019), and SAFN (Xu et al., 2019). For PDA, we consider PADA (Cao et al., 2018) and SAFN (Xu et al., 2019). For OPDA, we consider DANCE (Saito et al., 2020). For SFUDA, we consider SHOT (Liang et al., 2020) and DINE (Liang et al., 2021). For domain adaptive semantic segmentation, we consider AdaptSeg (Tsai et al., 2018) and AdvEnt (Vu et al., 2019). During selection, we explore 7 candidate values for each hyperparameter. Specifically, we select the loss coefficient for ATDOC, BNM, CDAN, PADA, SAFN, DANCE, SHOT, DINE, AdaptSeg, and AdvEnt, while the margin is selected for MDD and the temperature for MCC. Additionally, we perform two complex two-hyperparameters validation tasks. For classification, we tune the bottleneck dimension with 4 options in MCC and MDD, whereas for segmentation, we tune the training iteration with 8 options in AdaptSeg and AdvEnt. Detailed hyperparameter settings are provided in Appendix B.

**Implementation details.** We train UDA models using the Transfer Learning Library[1] on a single RTX TITAN 16GB GPU with a batch size of 32 and a total number of iterations of 5000. Unless specified, checkpoints are saved at the last iteration. ResNet-101 (He et al., 2016) is used for *VisDA* and segmentation tasks, ResNet-34 (He et al., 2016) for *DomainNet*, and ResNet-50 (He et al., 2016) for other benchmarks. Source-based validation methods allocate 80% of the source data for training and the remaining 20% for validation.

---

[1] https://github.com/thuml/Transfer-Learning-Library

Table 4: CDA accuracy (%) on *DomainNet-126*.

| Method | CDAN | | | | | BNM | | | | | ATDOC | | | | |
|--------|------|------|------|------|------|------|------|------|------|------|-------|------|------|------|------|
| | → C | → P | → R | → S | avg | → C | → P | → R | → S | avg | → C | → P | → R | → S | avg |
| Entropy | **67.09** | **65.80** | 74.42 | **59.34** | **66.66** | 63.36 | 64.28 | 74.31 | 48.69 | 62.66 | 63.75 | 61.85 | 79.60 | 52.17 | 64.34 |
| InfoMax | **67.09** | **65.80** | 74.42 | **59.34** | **66.66** | 67.05 | 64.28 | 74.31 | 55.67 | 65.33 | 63.75 | 61.85 | 79.60 | 52.17 | 64.34 |
| SND | **67.09** | 64.68 | 74.42 | **59.34** | 66.38 | 56.56 | 54.50 | 74.31 | 42.37 | 56.93 | 63.75 | 61.85 | 79.60 | 47.00 | 63.05 |
| Corr-C | 57.35 | 62.88 | 74.42 | 54.63 | 62.32 | 59.75 | 63.41 | **77.62** | 42.37 | 60.79 | 59.98 | 62.27 | 74.42 | 53.69 | 62.59 |
| EnsV | 65.88 | 65.27 | **74.44** | 57.42 | 65.75 | **67.86** | **66.06** | **77.62** | **57.69** | **67.31** | **70.30** | **68.44** | **80.01** | **61.73** | **70.12** |
| Worst | 57.35 | 60.76 | 73.44 | 51.41 | 60.74 | 55.79 | 54.50 | 74.31 | 42.37 | 56.74 | 59.98 | 61.85 | 74.42 | 47.00 | 60.81 |
| Best | 67.09 | 65.80 | 74.44 | 59.34 | 66.66 | 67.86 | 66.50 | 78.68 | 58.49 | 67.88 | 70.30 | 68.44 | 80.38 | 62.23 | 70.34 |

Table 5: PDA accuracy (%) on *Office-Home*.

| Method | PADA | | | | | SAFN | | | | |
|--------|------|------|------|------|------|------|------|------|------|------|
| | → Ar | → Cl | → Pr | → Re | avg | → Ar | → Cl | → Pr | → Re | avg |
| SourceRisk | 57.21 | 41.90 | 64.48 | 71.89 | 58.87 | 66.82 | 54.71 | 74.41 | 76.48 | 68.11 |
| IWCV | 59.65 | 50.51 | 66.84 | 72.96 | 62.49 | 69.36 | 53.91 | 71.78 | 76.38 | 67.86 |
| DEV | 66.88 | 49.29 | 72.40 | 70.46 | 64.76 | 69.36 | 54.94 | 73.95 | 76.06 | 68.58 |
| RV | 57.79 | 40.87 | 63.87 | 70.83 | 58.34 | 68.98 | 52.74 | 72.83 | 77.14 | 67.92 |
| Entropy | 60.08 | 46.51 | 53.16 | 62.47 | 55.56 | **71.75** | 55.62 | 76.36 | 76.59 | 70.08 |
| InfoMax | 60.08 | 51.40 | 60.20 | 66.67 | 59.59 | 63.67 | 51.74 | 69.64 | 73.62 | 64.67 |
| SND | 67.80 | 50.71 | 59.46 | 67.13 | 61.27 | **71.75** | 51.74 | 76.36 | 78.36 | 69.55 |
| Corr-C | 61.34 | 45.65 | 54.90 | 62.25 | 56.04 | 71.23 | 55.70 | **76.94** | **79.13** | **70.75** |
| EnsV | **68.54** | **55.60** | **69.86** | **78.23** | **68.06** | 70.98 | **56.12** | 75.67 | 78.48 | 70.31 |
| Worst | 56.29 | 39.76 | 50.49 | 59.31 | 51.46 | 62.48 | 49.91 | 68.50 | 73.62 | 63.63 |
| Best | 69.33 | 55.86 | 74.55 | 79.59 | 69.83 | 73.37 | 58.09 | 77.35 | 79.33 | 72.03 |

Table 8: OPDA H-score (%) on *Office-Home*. SFUDA accuracy (%) on *Office-31* and *VisDA*.

| Method | DANCE | | | | | SHOT | | | | *DINE* |
|--------|-------|------|------|------|------|------|------|------|------|------|
| | →Ar | →Cl | →Pr | →Re | avg | →A | →D | →W | avg | T→V |
| Entropy | 32.00 | 39.48 | 27.52 | 38.08 | 34.27 | 71.67 | 90.76 | 88.68 | 83.70 | 71.99 |
| InfoMax | 32.00 | 39.48 | 27.52 | 38.01 | 34.25 | 71.67 | 90.76 | 88.68 | 83.70 | 71.99 |
| SND | 15.05 | 4.33 | 23.75 | 16.79 | 14.98 | 71.67 | 90.76 | 88.68 | 83.70 | **74.43** |
| Corr-C | 29.60 | 4.33 | 23.75 | 16.79 | 18.62 | 71.58 | 90.76 | 90.19 | 84.18 | 71.99 |
| EnsV | **77.01** | **51.36** | **78.81** | **68.65** | **68.96** | **74.85** | **94.78** | **91.82** | **87.15** | **74.43** |
| Worst | 15.05 | 4.33 | 15.17 | 16.79 | 12.84 | 71.56 | 90.76 | 88.68 | 83.67 | 71.99 |
| Best | 77.01 | 66.29 | 78.81 | 69.81 | 72.98 | 75.06 | 94.78 | 93.33 | 87.72 | 76.17 |

## 4.2 RESULTS

We evaluate the validation performance of EnsV in 5 UDA scenarios: CDA, PDA, OPDA, SFUDA (classification), and CDA (semantic segmentation). The results are averaged using three random seeds. We present averaged results for UDA tasks with the same target domain. The 'Worst' selection indicates the one with the lowest performance, while the 'Best' selection indicates the opposite.

**CDA**: We provide comprehensive model selection results for 6 typical UDA methods on *Office-Home*, *Office-31*, and *VisDA* in Tables 2 and 3. EnsV methods consistently outperform other validation methods in terms of the average accuracy on each benchmark and furthermore, consistently achieve near-best selections. For results on *DomainNet-126*, we report them in Table 4. EnsV consistently makes selections above the median, while other approaches exhibit more variability.

**PDA**: For partial-set UDA with label shift, we perform hyperparameter selections for two typical UDA methods on *Office-Home* (Table 5). Our EnsV outperforms all other model selection methods by a significant margin in terms of average accuracy.

**OPDA**: In the open-partial-set UDA with label shift, we choose a representative method DANCE for validation on *Office-Home* (Table 8). Prior model selection works have not explored this challenging setting, resulting in poor selections. However, our EnsV achieves selections close to the best.

**SFUDA**: For source-free UDA (SFUDA), we choose SHOT for the white-box setting on *Office-31* and DINE for the black-box setting on *VisDA* (Table 8). EnsV consistently maintains near-best selections, while other target-based approaches occasionally make near-worst selections.

**Validation with two hyperparameters**: We conduct practical two-hyperparameters model selection experiments on classification tasks (Table 9) and segmentation tasks (Table 7). Most model selection

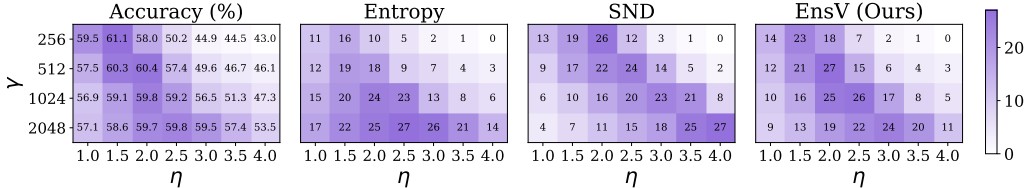

Figure 2: Qualitative comparisons of two-hyperparameters validation for MCC on Ar → Cl.

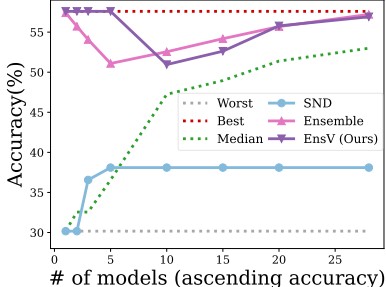

Figure 3: MDD on Ar→Cl.

Table 6: ViT results.

| Method | BNM |
|---|---|
| Entropy | 28.21 |
| InfoMax | 28.21 |
| SND | 52.42 |
| Corr-C | 28.21 |
| EnsV | **55.16** |
| Worst | 28.21 |
| Best | 55.16 |

Table 7: Segmentation mIoU (%).

| Method | AdaptSegt | AdvEnt |
|---|---|---|
| SourceRisk | 39.52 | 39.08 |
| Entropy | 39.47 | 38.41 |
| SND | **40.69** | 40.02 |
| EnsV | **40.69** | **40.67** |
| Worst | 35.32 | 34.22 |
| Best | 42.20 | 41.78 |

studies focus on classification, with limited attention (Saito et al., 2021) to segmentation. We find EnsV consistently achieves near-optimal selections on both tasks, outperforming other generic methods like Entropy and SND.

### 4.3 ANALYSIS

**Qualitative comparison.** We perform a qualitative comparison between two state-of-the-art target-based model selection methods, Entropy and SND, and our EnsV. In Figure 2, we present the rankings of the 28 candidate checkpoints in ascending order based on the respective selection metric of each approach. On the left side, we show the rankings according to the real target accuracy and denote the accuracy for each candidate model. Our EnsV demonstrates a high level of consistency with the real target accuracy, while the other methods exhibit significant deviations. This highlights the superior reliability of our EnsV over other methods.

**Robustness to architectures.** Architecture plays a significant role in the ensemble. In our experiments, we assess the effectiveness of EnsV using various ResNet backbone variants and observe consistent success across different scales. For further study, we conduct validation experiments using the ViT-B (Dosovitskiy et al., 2021) architecture on the R→S task with BNM. The validation results, presented in Table 6, demonstrate that EnsV achieves the best selection. However, all other target-based methods except SND make the worst selection.

**Performance of role models.** The effectiveness of our ensemble-based validation method, EnsV, relies on the performance of the role model. We evaluate the target performance of role models for various UDA methods in 4 UDA settings on *Office-Home* and present the results in Table 10. Through a comparison of ensemble performance with model selection performance in our empirical experiments, we demonstrate that the ensemble consistently exhibits high performance. The success of EnsV can be attributed to the robust role model provided by the ensemble. We present the results of the weight-based ensemble (Wortsman et al., 2022), denoted as 'W-Avg,' and the EnsV variant based on this ensemble, denoted as 'EnsV-W.' While the weight ensemble also shows competitiveness, it necessitates all candidate models to have the same architecture. Thus, we recommend the simple and generic prediction-based ensemble. Kindly refer to Appendix D for full results.

**Robustness to bad candidates.** The robustness of deep ensembling to bad checkpoints is critical for its effectiveness. We conduct two-hyperparameter validation experiments using MDD on Ar→Cl to assess this. In the worst-case scenario where we have only one good checkpoint and several bad checkpoints, the ensemble results may be heavily influenced by the bad checkpoints, leading to poor selections. To analyze this, we rank the 28 candidate checkpoints based on their true target accuracy.

Table 9: Two-hyperparameter validation accuracy (%) on *Office-Home*.

| Method | MDD | | | | | MCC | | | | | *Home* |
|---|---|---|---|---|---|---|---|---|---|---|---|
| | Ar → Cl | Cl → Pr | Pr → Re | Re → Ar | avg | Ar → Cl | Cl → Pr | Pr → Re | Re → Ar | avg | AVG |
| SourceRisk | 55.99 | 73.15 | 78.77 | 69.39 | 69.33 | 57.91 | 76.84 | 81.13 | 72.89 | 72.19 | 70.76 |
| IWCV | 37.89 | 72.92 | 80.42 | 58.43 | 62.42 | 46.09 | 77.74 | 80.68 | 74.45 | 69.74 | 66.08 |
| DEV | 52.60 | 72.11 | 53.36 | 67.70 | 61.44 | 59.47 | 76.84 | 81.94 | 74.08 | 73.08 | 67.26 |
| RV | **57.59** | 72.25 | 80.83 | 70.79 | 70.37 | 59.13 | 76.84 | 82.03 | 71.98 | 72.50 | 71.44 |
| Entropy | 57.21 | **73.19** | 80.06 | **72.31** | 70.69 | 59.75 | 77.77 | 82.37 | 74.33 | 73.56 | 72.13 |
| InfoMax | **57.59** | 72.92 | 80.06 | **72.31** | **70.72** | 59.70 | **78.73** | **82.58** | 70.33 | 72.84 | 71.78 |
| SND | 38.10 | 56.45 | 70.03 | 65.10 | 57.42 | 53.49 | 74.97 | 77.25 | 74.12 | 69.96 | 63.69 |
| Corr-C | 30.17 | 44.74 | 57.15 | 50.76 | 45.71 | 44.90 | 56.75 | 74.32 | 67.61 | 60.90 | 53.31 |
| EnsV | 56.91 | 72.74 | **80.93** | 71.16 | 70.44 | **60.39** | 78.71 | 82.28 | **74.91** | **74.07** | **72.26** |
| Worst | 30.17 | 39.81 | 53.36 | 50.76 | 43.53 | 43.02 | 56.75 | 73.47 | 67.24 | 60.12 | 51.83 |
| Best | 57.59 | 73.35 | 80.93 | 72.52 | 71.10 | 61.10 | 78.94 | 83.04 | 75.36 | 74.61 | 72.86 |

Table 10: Accuracy (%) of the ensemble on *Office-Home*.

| Method | CDA | | | | | | PDA | | OPDA | SFUDA |
|---|---|---|---|---|---|---|---|---|---|---|
| | ATDOC | BNM | CDAN | MCC | MDD | SAFN | PADA | SAFN | DANCE | SHOT |
| W-Avg | 72.04 | 70.48 | 69.30 | 72.77 | 69.39 | 66.65 | 67.46 | 70.11 | 64.97 | 71.82 |
| Ensemble | **72.13** | 70.86 | **70.32** | **72.82** | **69.80** | 67.12 | **68.23** | 70.71 | **69.31** | **71.94** |
| EnsV-W | 71.72 | 70.74 | 69.81 | 72.70 | 69.23 | **67.38** | 68.21 | **71.91** | 66.85 | 71.74 |
| EnsV | 71.74 | **70.90** | 70.25 | 72.65 | 69.60 | 67.04 | 68.06 | 70.71 | 68.96 | 71.88 |
| Worst | 65.26 | 60.75 | 56.86 | 67.93 | 50.88 | 59.88 | 51.46 | 63.63 | 12.84 | 67.21 |
| Best | 72.04 | 71.11 | 70.43 | 73.14 | 70.04 | 67.55 | 69.83 | 72.03 | 72.98 | 72.05 |

Starting with the best and worst checkpoints, we gradually introduce more bad checkpoints into the ensemble. By observing the ensembling and validation performance in Figure 3, we study the impact of bad checkpoints. Despite the presence of bad checkpoints, both the prediction-average Ensemble and our EnsV consistently prioritize selections above the median, demonstrating their resilience. In contrast, the state-of-the-art method SND falls short in surpassing the median selection.

## 5 DISCUSSIONS

**Limitations.** In unsupervised model selection, EnsV reliably avoids the worst selection but may face low-performance challenges in two scenarios: (i) Selecting the optimal candidate from a pool where most options perform poorly, and (ii) Choosing between a single poor model and a good one.

**Key insights for model selection in UDA.** After conducting a thorough comparison of existing model selection methods, we have identified the following key insights regarding the model selection problem in UDA:

- Firstly, we emphasize the importance of addressing model selection in UDA, which has often been overlooked. We call for increased attention (You et al., 2019; Saito et al., 2021; Musgrave et al., 2022) and reporting of validation methods used to determine hyperparameters, rather than relying solely on fixed hyperparameters or limited hyperparameter analyses within a predefined range.

- Secondly, among existing methods, we recommend the reverse validation (RV) approach as the most reliable method for CDA when source data is accessible. However, it requires additional model re-training, making it less lightweight compared to target-based validation methods. It is worth noting that no existing validation methods can effectively handle the validation of UDA models in diverse scenarios with varying domain shifts or tasks.

- Lastly, our ensemble-based validation method, EnsV, demonstrates superior performance across various UDA scenarios without any instances of poor selection let alone worst selection. EnsV is a post-hoc method that leverages the ensemble of available candidate models, eliminating the need for additional model training. We suggest employing EnsV as a simple, versatile, yet highly stable approach for model selection or hyperparameter selection in UDA study. Furthermore, we believe that EnsV has the potential to offer valuable insights for model selection in a wide range of applications beyond UDA.

## REPRODUCIBILITY STATEMENT

We propose an ensemble-based method EnsV for unsupervised model selection in domain adaptation. Our method is frustratingly easy, without any hyperparameters or tricks and anyone interested can breezily implement our method. We plan to release a full implementation upon acceptance.

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

## A    PROOF OF PROPOSITION 1

Given the use of negative log-likelihood (NLL) as the loss function, defined as $l(p, y) = -\log p^y$. We first prove the first inequality, i.e., $l(\frac{1}{m}\sum_{i=1}^m f(x, \theta_i), y) < \frac{1}{m}\sum_{i=1}^m l(f(x, \theta_i), y)$. We employ Jensen's inequality, which asserts that for a real-valued, convex function $\varphi$ with its domain as a subset of $\mathbb{R}$ and numbers $t_1, \ldots, t_n$ in its domain, the inequality $\varphi\left(\frac{1}{n}\sum_{i=1}^n t_i\right) \le \frac{1}{n}\sum_{i=1}^n \varphi(t_i)$ holds. Given that $-\log$ is a convex function, and in the main text, we assume $m$ is greater than 1, and candidate models have different weights $\theta$, resulting in different discriminative mappings of $f(x, \theta)$. we can obtain $l(\frac{1}{m}\sum_{i=1}^m f(x, \theta_i), y) < \frac{1}{m}\sum_{i=1}^m l(f(x, \theta_i), y)$, without the equal situation.

Then for the proof of the second inequality $\frac{1}{m}\sum_{i=1}^m l(f(x, \theta_i), y) < l(f(x, \theta_{\text{worst}}), y)$, we leverage the transitivity of inequalities. Because $\theta_{\text{worst}}$ denotes the worst candidate model, for any other candidate model $\theta_i$, we have $l(f(x, \theta_{\text{worst}}), y) < l(f(x, \theta_i), y)$. Because $-\log$ is a monotonically decreasing function, we can surely have $\frac{1}{m}\sum_{i=1}^m l(f(x, \theta_i), y) < \frac{1}{m}\sum_{i=1}^m l(f(x, \theta_{\text{worst}}), y)$, i.e., $\frac{1}{m}\sum_{i=1}^m l(f(x, \theta_i), y) < l(f(x, \theta_{\text{worst}}), y)$.

## B    HYPERPARAMETER CONFIGURATIONS

In our experiments, we adopt the setting of previous studies (You et al., 2019; Saito et al., 2021) by tuning a single hyperparameter for various UDA methods. The comprehensive hyperparameter settings can be found in Table 11. For MCC (Jin et al., 2020) and MDD (Zhang et al., 2019), we also explore different bottleneck dimensions: $256, 512, 1024, 2048$. Additionally, in semantic segmentation tasks, we consider the training iteration following SND (Saito et al., 2021).

Table 11: Overview of the UDA methods validated and their associated hyperparameters

| UDA method | UDA Type | Hyperparameter | Search Space | Default Value |
|---|---|---|---|---|
| ATDOC (Liang et al., 2021) | closed-set self-training | loss coefficient $\lambda$ | $\{0.02, 0.05, 0.1,$ $0.2, 0.5, 1.0, 2.0\}$ | 0.2 |
| BNM (Cui et al., 2020) | closed-set output regularization | loss coefficient $\lambda$ | $\{0.02, 0.05, 0.1,$ $0.2, 0.5, 1.0, 2.0\}$ | 1.0 |
| CDAN (Long et al., 2018) | closed-set feature alignment | loss coefficient $\lambda$ | $\{0.05, 0.1, 0.2,$ $0.5, 1.0, 2.0, 5.0\}$ | 1.0 |
| MCC (Jin et al., 2020) | closed-set output regularization | temperature $T$ | $\{1.0, 1.5, 2.0,$ $2.5, 3.0, 3.5, 4.0\}$ | 2.5 |
| MDD (Zhang et al., 2019) | closed-set output alignment | margin factor $\gamma$ | $\{0.5, 1.0, 2.0,$ $3.0, 4.0, 5.0, 6.0\}$ | 4.0 |
| SAFN (Xu et al., 2019) | closed/partial-set feature regularization | loss coefficient $\lambda$ | $\{0.002, 0.005, 0.01,$ $0.02, 0.05, 0.1, 0.2\}$ | 0.05 |
| PADA (Cao et al., 2018) | partial-set feature alignment | loss coefficient $\lambda$ | $\{0.05, 0.1, 0.2,$ $0.5, 1.0, 2.0, 5.0\}$ | 1.0 |
| DANCE (Saito et al., 2020) | open-partial-set self-supervision | loss coefficient $\eta$ | $\{0.02, 0.05, 0.1,$ $0.2, 0.5, 1.0, 2.0\}$ | 0.05 |
| SHOT (Liang et al., 2020) | source-free hypothesis transfer | loss coefficient $\beta$ | $\{0.03, 0.05, 0.1,$ $0.3, 0.5, 1.0, 3.0\}$ | 0.3 |
| DINE (Liang et al., 2021) | black-box knowledge distillation | loss coefficient $\beta$ | $\{0.05, 0.1, 0.2,$ $0.5, 1.0, 2.0, 5.0\}$ | 1.0 |
| AdaptSeg (Tsai et al., 2018) | closed-set output alignment | loss coefficient $\lambda$ | $\{0.0001, 0.0003, 0.001,$ $0.003, 0.01, 0.03\}$ | 0.0002 |
| AdvEnt (Vu et al., 2019) | closed-set output alignment | loss coefficient $\lambda$ | $\{0.0001, 0.0003, 0.001,$ $0.003, 0.01, 0.03\}$ | 0.001 |

## C    MODEL SELECTION BASELINES

Let $\{p_t^i\}_{i=1}^{n_t}$ represent the target probability output, and let $P \in \mathbb{R}^{n_t \times C}$ denote the prediction matrix. We provide a brief overview of the existing model selection approaches.

**Source risk.** The Source risk approach (SourceRisk) (Ganin & Lempitsky, 2015) utilizes a hold-out source validation set to select the model $\theta_k$ with the best performance on this set as the final choice. However, this method is limited in its ability to handle significant domain shifts between domains and introduces additional hyperparameters during the splitting of the validation set.

**Importance-weighted source risk.** Directly taking source risk as target risk is unreliable due to domain distribution shifts between domains. To address this challenge, Sugiyama et al. (2007) propose Importance-Weighted Cross Validation (IWCV), which re-weights the source risk using a source-target density ratio estimated in the input space. You et al. (2019) further enhance IWCV by introducing Deep Embedded Validation (DEV), which estimates the density ratio in the feature space using a domain discriminator and controls the variance. Both IWCV and DEV rely on the importance weighting technique (Cortes et al., 2008), which assumes that the target distribution is included in the source distribution (Sugiyama et al., 2007), making the weighting unreliable in scenarios with significant covariate shift and label shift. In addition, both IWCV and DEV involve hyperparameters and extra model training during the density ratio estimation process.

**Reversed source risk.** Building upon the concept of reverse cross-validation (Zhong et al., 2010), Ganin et al. (2016) propose a novel Reverse Validation approach (RV). This method first conducts source-to-target adaptation to obtain a UDA model, which enables the acquisition of pseudo labels for the target unlabeled data. Subsequently, Reverse Validation performs a reversed adaptation from the pseudo-labeled target to the source and utilizes the source risk in this reversed adaptation task for validation. Reverse Validation heavily relies on the symmetry between domains and is unable to handle label shift. Additionally, this approach involves hyperparameters for dataset splitting.

**Entropy.** Morerio et al. (2017) propose using the mean Shannon's Entropy of target predictions as a validation metric, which encourages confident predictions. The motivation behind this is that the decision boundary should avoid crossing high-density regions in the target structure (Grandvalet & Bengio, 2004; Chapelle & Zien, 2005). Lower Entropy scores indicate better model performance for this metric.

$$\text{Entropy} = -\frac{1}{n_\text{t}} \sum_{i=1}^{n_\text{t}} \sum_{j=1}^{C} P_{ij} \log P_{ij}, \qquad \text{InfoMax} = -\sum_{j=1}^{C} \bar{p} \log \bar{p} + \frac{1}{n_\text{t}} \sum_{i=1}^{n_\text{t}} \sum_{j=1}^{C} P_{ij} \log P_{ij}$$

**Information maximization.** The Entropy score only considers sample-wise certainty, which can be misleading when confident predictions are biased towards a small fraction of classes (Saito et al., 2021). To address this challenge, Musgrave et al. (2022) utilize input-output mutual information maximization (InfoMax) (Bridle et al., 1991) as a validation metric. In contrast to Entropy, InfoMax includes an additional class-balance regularization by encouraging the averaged prediction $\bar{p} = \frac{1}{n_\text{t}} \sum_{i=1}^{n_\text{t}} P_{ij}, \quad \bar{p} \in \mathbb{R}^C$ to have a large entropy. Higher InfoMax scores indicate better model performance according to this metric.

**Neighborhood consistency.** Saito et al. (2021) introduce Soft Neighborhood Density (SND), a novel metric that focuses on neighborhood consistency. SND leverages softmax predictions as features and constructs a sample-sample similarity matrix. This matrix is transformed into a probabilistic distribution using the softmax function: $S = \text{softmax}(PP^T/\tau), \quad S \in \mathbb{R}^{n_\text{t} \times n_\text{t}}$. Here, $\tau$ is a small temperature parameter that sharpens the distribution, enabling the differentiation between nearby and distant samples. SND promotes high neighborhood consistency by encouraging samples to have similar predictions to other points in their neighborhood, resulting in larger SND scores.

$$\text{SND} = -\frac{1}{n_\text{t}} \sum_{i=1}^{n_\text{t}} \sum_{j=1}^{n_\text{t}} S_{ij} \log S_{ij}, \qquad \text{Corr-C} = \frac{\text{sum}(\text{diag}(P^T P))}{\|P^T P\|_\text{F}}$$

**Class correlation.** Tu et al. (2023) introduce Corr-C, a class correlation-based metric that evaluates class diversity and prediction certainty simultaneously. Corr-C calculates the cosine similarity between the class correlation matrix and an identity matrix. Lower Corr-C scores are indicative of better model performance based on this metric.

## D  FULL MODEL SELECTION RESULTS

Due to space constraints in the main text, we have presented the average results for tasks with the same target domain. For example, in the case of the *Office-Home* dataset, UDA tasks including 'Cl→Ar', 'Pr→Ar' and 'Re→Ar' share the common target domain 'Ar.' As a result, we have averaged

the results of these three UDA tasks and reported the averaged value in the tables within our main text under the row labeled '→ Ar'.

Furthermore, it's important to distinguish between the 'avg' row, which signifies the average results within each UDA method's rows to the left of the 'avg' row, and the 'AVG' row, which represents the averaged results across all 'avg' rows associated with different UDA methods. Consequently, the 'AVG' row can be considered more reliable and representative for drawing conclusions.

In our evaluation, we conduct hyperparameter selection for both classification and segmentation tasks. For open-set experiments, we utilize the H-score (%) (Fu et al., 2020; Bucci et al., 2020) metric, which combines the accuracy of known classes and unknown samples. For semantic segmentation tasks, we employ the mean intersection-over-union (mIoU) (%) (Tsai et al., 2018; Vu et al., 2019) metric. For all other classification tasks, we measure the accuracy (%). Due to space constraints in the main text, we consolidate the results of UDA tasks with the same target domain. Please refer to Table 12 to Table 26 for the complete set of validation results.

Table 12: Accuracy (%) of a closed-set UDA method ATDOC (Liang et al., 2021) on *Office-Home*.

| Method | Ar → Cl | Ar → Pr | Ar → Re | Cl → Ar | Cl → Pr | Cl → Re | Pr → Ar | Pr → Cl | Pr → Re | Re → Ar | Re → Cl | Re → Pr | AVG |
|---|---|---|---|---|---|---|---|---|---|---|---|---|---|
| SourceRisk | 51.41 | **77.31** | 78.17 | **66.87** | 74.36 | 75.60 | 61.85 | 48.04 | 76.06 | 71.16 | 58.14 | **84.05** | 68.59 |
| IWCV | 55.88 | 76.57 | 78.88 | 66.25 | 74.50 | 78.33 | 65.60 | 48.04 | 80.58 | **72.06** | 58.14 | 83.87 | 69.89 |
| DEV | 51.41 | 76.55 | 78.88 | 66.25 | 74.36 | 77.67 | 64.77 | 51.29 | 81.62 | 71.16 | **59.98** | 82.43 | 69.70 |
| RV | 56.38 | 76.12 | 80.01 | 66.25 | 76.80 | 78.33 | 67.82 | 55.62 | 80.58 | 71.98 | 56.40 | 83.87 | 70.85 |
| Entropy | 55.88 | 74.14 | 78.88 | 59.25 | 74.52 | 77.67 | 64.19 | 54.39 | 78.54 | 67.57 | 57.23 | 80.96 | 68.60 |
| InfoMax | 55.88 | 74.14 | 78.88 | 59.25 | 77.74 | 77.67 | 64.19 | 54.39 | 78.54 | 67.57 | 56.61 | 80.96 | 68.82 |
| SND | 55.88 | 74.14 | 78.88 | 59.25 | 74.52 | 75.21 | 64.19 | 54.39 | 78.54 | 67.57 | 56.61 | 80.96 | 68.34 |
| Corr-C | 51.41 | 72.00 | 76.04 | 59.37 | 69.36 | 69.54 | 61.85 | 48.04 | 76.06 | 69.30 | 51.71 | 80.31 | 65.42 |
| EnsV-W | **57.85** | 76.57 | **81.04** | 66.25 | **79.48** | **78.52** | 67.94 | 55.62 | **82.17** | 71.9 | 59.24 | 84.03 | 71.72 |
| EnsV | **57.85** | 76.57 | 80.54 | 66.25 | 78.82 | **78.52** | 67.94 | 57.07 | **82.17** | 71.9 | 59.24 | 84.03 | **71.74** |
| Worst | 51.41 | 72.00 | 76.04 | 59.25 | 69.36 | 69.54 | 61.85 | 48.04 | 76.06 | 67.57 | 51.71 | 80.31 | 65.26 |
| Best | 58.01 | 77.31 | 81.04 | 66.91 | 79.48 | 78.52 | 67.94 | 57.07 | 82.17 | 72.06 | 59.98 | 84.03 | 72.04 |

Table 13: Accuracy (%) of a closed-set UDA method BNM (Cui et al., 2020) on *Office-Home*.

| Method | Ar → Cl | Ar → Pr | Ar → Re | Cl → Ar | Cl → Pr | Cl → Re | Pr → Ar | Pr → Cl | Pr → Re | Re → Ar | Re → Cl | Re → Pr | AVG |
|---|---|---|---|---|---|---|---|---|---|---|---|---|---|
| SourceRisk | 56.93 | **77.00** | 77.74 | 57.64 | **73.33** | 69.36 | 56.45 | 42.38 | 77.19 | 73.22 | 52.90 | 82.26 | 66.37 |
| IWCV | 46.46 | **77.00** | 79.30 | 63.86 | 61.34 | 62.54 | 63.95 | 42.38 | 78.01 | 71.86 | 55.65 | 83.92 | 65.52 |
| DEV | 57.75 | 71.62 | 79.30 | 57.64 | 67.90 | 75.46 | **66.21** | **54.04** | 78.01 | **73.42** | 57.37 | 82.25 | 68.41 |
| RV | **58.67** | **77.00** | 79.30 | 65.68 | **73.33** | 75.46 | 65.64 | 52.05 | **81.25** | **73.42** | **59.54** | 83.92 | 70.44 |
| Entropy | 53.40 | 67.04 | 78.04 | 63.41 | 71.44 | 73.93 | 63.58 | 52.69 | 80.95 | 71.86 | 57.37 | **83.96** | 68.14 |
| InfoMax | 53.40 | 67.04 | 78.04 | 63.41 | 71.44 | 73.93 | 63.58 | 52.69 | 80.95 | 71.86 | 57.37 | **83.96** | 68.14 |
| SND | 53.40 | 67.04 | 78.04 | 63.41 | 71.44 | 73.93 | 63.58 | 52.69 | 80.95 | 71.86 | 57.37 | **83.96** | 68.14 |
| Corr-C | 46.46 | 67.04 | 74.82 | 49.73 | 61.34 | 62.54 | 56.45 | 42.38 | 74.41 | 68.11 | 47.26 | 78.51 | 60.76 |
| EnsV-W | **58.67** | **77.00** | 80.61 | 66.21 | **73.33** | **76.75** | **66.21** | 53.93 | **81.25** | **73.42** | 57.59 | 83.92 | 70.74 |
| EnsV | **58.67** | **77.00** | 80.61 | 66.21 | **73.33** | **76.75** | **66.21** | 53.93 | **81.25** | **73.42** | **59.54** | 83.92 | **70.90** |
| Worst | 46.46 | 67.04 | 74.82 | 49.73 | 61.34 | 62.54 | 56.45 | 42.38 | 74.41 | 68.11 | 47.26 | 78.51 | 60.75 |
| Best | 58.67 | 77.00 | 80.61 | 67.16 | 74.16 | 76.75 | 66.21 | 54.04 | 81.36 | 73.42 | 59.82 | 84.12 | 71.11 |

Table 14: Accuracy (%) of a closed-set UDA method CDAN (Long et al., 2018) on *Office-Home*.

| Method | Ar → Cl | Ar → Pr | Ar → Re | Cl → Ar | Cl → Pr | Cl → Re | Pr → Ar | Pr → Cl | Pr → Re | Re → Ar | Re → Cl | Re → Pr | AVG |
|---|---|---|---|---|---|---|---|---|---|---|---|---|---|
| SourceRisk | 43.41 | 62.51 | 75.51 | 43.96 | 61.59 | 57.70 | 53.75 | 37.50 | 73.22 | 67.28 | 47.01 | 84.39 | 58.99 |
| IWCV | 43.14 | 62.51 | 77.81 | 54.58 | 56.14 | 65.14 | 37.50 | **81.85** | 74.08 | 43.02 | 84.39 | 60.41 |
| DEV | 57.16 | 71.75 | 77.81 | 62.46 | 55.64 | 71.08 | 65.14 | 56.54 | **81.85** | 74.08 | 57.43 | 78.89 | 67.49 |
| RV | 57.16 | 71.75 | 77.78 | 63.62 | 72.92 | 73.40 | 65.14 | 54.50 | **81.85** | **74.21** | **58.56** | 83.37 | 69.52 |
| Entropy | **57.55** | 72.43 | 77.74 | 63.62 | 72.92 | 73.40 | **65.27** | 56.66 | 81.20 | 74.08 | 58.47 | 83.76 | 69.76 |
| InfoMax | **57.55** | 72.43 | 77.74 | 63.62 | 72.92 | 73.40 | **65.27** | 56.66 | 81.20 | 74.08 | 58.47 | 83.76 | 69.76 |
| SND | **57.55** | 72.43 | 77.78 | **64.61** | 73.73 | 73.40 | 65.14 | 56.66 | **81.85** | 74.08 | 58.47 | **84.73** | 70.04 |
| Corr-C | 43.14 | 63.05 | 73.61 | 43.96 | 54.58 | 56.12 | 51.75 | 37.50 | 73.22 | 65.80 | 43.00 | 77.25 | 56.91 |
| EnsV-W | 57.18 | 73.30 | 77.78 | 63.37 | 73.89 | 73.38 | 65.14 | 55.44 | 81.36 | 73.88 | **58.56** | 84.39 | 69.81 |
| EnsV | **57.55** | **73.71** | **78.33** | **64.61** | **73.73** | **74.39** | 65.14 | 56.66 | **81.85** | 73.88 | **58.56** | **84.73** | **70.25** |
| Worst | 43.14 | 62.51 | 73.61 | 43.96 | 54.58 | 56.12 | 51.63 | 37.50 | 73.22 | 65.80 | 43.00 | 77.25 | 56.86 |
| Best | 57.55 | 73.71 | 78.33 | 64.61 | 73.89 | 74.39 | 65.76 | 56.66 | 81.85 | 74.21 | 59.50 | 84.73 | 70.43 |

Table 15: Accuracy (%) of a closed-set UDA method MCC (Jin et al., 2020) on *Office-Home*.

| Method | Ar→Cl | Ar→Pr | Ar→Re | Cl→Ar | Cl→Pr | Cl→Re | Pr→Ar | Pr→Cl | Pr→Re | Re→Ar | Re→Cl | Re→Pr | AVG |
|---|---|---|---|---|---|---|---|---|---|---|---|---|---|
| SourceRisk | 57.23 | 78.19 | **81.75** | 60.65 | 76.50 | **78.79** | 64.15 | 53.15 | 82.17 | **74.91** | 59.20 | 83.96 | 70.89 |
| IWCV | **60.02** | 78.15 | 81.34 | 68.73 | **78.51** | 77.85 | 64.15 | **57.85** | 81.04 | 73.18 | 58.92 | 84.46 | 72.02 |
| DEV | 57.16 | 78.15 | 81.34 | **69.10** | 73.01 | 76.80 | 64.15 | **57.85** | 82.17 | 73.18 | 59.20 | 84.46 | 71.38 |
| RV | 59.34 | **78.53** | 80.70 | **69.10** | 77.83 | 78.22 | **67.20** | **57.85** | 82.24 | **74.91** | 59.20 | **85.54** | 72.56 |
| Entropy | 59.31 | **78.53** | 81.59 | 66.87 | 77.83 | **78.79** | **67.20** | **57.85** | **82.51** | 73.79 | 60.82 | **85.54** | 72.55 |
| InfoMax | **60.02** | 74.66 | **81.75** | 64.98 | 78.24 | 78.49 | 64.15 | 54.52 | 82.19 | 70.62 | 60.89 | 84.46 | 71.25 |
| SND | 53.56 | 77.43 | 79.46 | 67.28 | 76.48 | 76.80 | 65.06 | 54.34 | 81.04 | 74.82 | 58.92 | 85.24 | 70.87 |
| Corr-C | 53.56 | 77.43 | 79.46 | 67.28 | 76.48 | 76.80 | 65.06 | 54.34 | 81.04 | 74.82 | 58.92 | 85.24 | 70.87 |
| EnsV-W | 59.31 | 77.86 | 81.59 | **69.10** | **78.51** | **78.79** | 66.87 | **57.85** | 82.19 | 73.79 | **61.35** | 85.22 | **72.70** |
| EnsV | 59.31 | 77.86 | 81.59 | **69.10** | 77.83 | **78.79** | 66.87 | **57.85** | 82.19 | 73.79 | **61.35** | 85.22 | 72.65 |
| Worst | 53.56 | 73.44 | 79.25 | 60.65 | 73.01 | 75.76 | 59.74 | 53.15 | 79.55 | 67.78 | 57.18 | 82.11 | 67.93 |
| Best | 60.02 | 78.53 | 81.75 | 69.22 | 78.51 | 78.79 | 67.90 | 58.49 | 82.51 | 74.91 | 61.35 | 85.74 | 73.14 |

Table 16: Accuracy (%) of a closed-set UDA method MDD (Zhang et al., 2019) on *Office-Home*.

| Method | Ar→Cl | Ar→Pr | Ar→Re | Cl→Ar | Cl→Pr | Cl→Re | Pr→Ar | Pr→Cl | Pr→Re | Re→Ar | Re→Cl | Re→Pr | AVG |
|---|---|---|---|---|---|---|---|---|---|---|---|---|---|
| SourceRisk | 54.85 | 73.35 | 77.05 | 58.76 | 69.95 | 72.23 | 60.03 | 51.02 | 77.36 | 68.81 | 57.42 | 82.50 | 66.94 |
| IWCV | 56.40 | 69.52 | 76.59 | 58.76 | 67.40 | 69.43 | 61.89 | 56.43 | 76.82 | 71.94 | 56.68 | **84.43** | 67.19 |
| DEV | 57.71 | **75.42** | 77.05 | 58.76 | **72.99** | 70.51 | **63.95** | 56.43 | 80.26 | 70.54 | 56.68 | 82.14 | 68.54 |
| RV | **58.05** | **75.42** | 76.59 | 63.54 | 69.95 | **73.74** | **63.95** | 51.02 | **80.38** | 72.23 | 58.17 | **84.43** | 68.96 |
| Entropy | 57.73 | 74.54 | **78.22** | **64.07** | **72.99** | **73.74** | **63.95** | 55.85 | **80.38** | 71.61 | 59.31 | 84.28 | 69.72 |
| InfoMax | **58.05** | 74.54 | **78.22** | **64.07** | **72.99** | **73.74** | **63.95** | 55.85 | **80.38** | 71.61 | 59.31 | 84.28 | **69.75** |
| SND | **58.05** | **75.42** | 77.05 | 44.99 | **72.99** | 48.06 | 37.08 | 21.60 | 80.26 | 71.94 | 34.39 | **84.43** | 58.86 |
| Corr-C | 39.08 | 59.74 | 69.61 | 44.99 | 54.58 | 48.06 | 37.08 | 21.60 | 64.22 | 61.31 | 34.39 | 75.87 | 50.88 |
| EnsV-W | 54.89 | **75.42** | 78.01 | 61.89 | **72.99** | 72.23 | 63.08 | 56.43 | 79.66 | **72.23** | 60.02 | 83.96 | 69.23 |
| EnsV | 56.40 | **75.42** | 77.05 | **64.07** | **72.99** | 72.23 | 63.08 | **57.02** | 80.26 | **72.23** | 60.02 | **84.43** | 69.60 |
| Worst | 39.08 | 59.74 | 69.61 | 44.99 | 54.58 | 48.06 | 37.08 | 21.60 | 64.22 | 61.31 | 34.39 | 75.87 | 50.88 |
| Best | 58.05 | 75.42 | 78.22 | 64.07 | 72.99 | 73.74 | 63.95 | 57.02 | 80.38 | 72.23 | 60.02 | 84.43 | 70.04 |

Table 17: Accuracy (%) of a closed-set UDA method SAFN (Xu et al., 2019) on *Office-Home*.

| Method | Ar→Cl | Ar→Pr | Ar→Re | Cl→Ar | Cl→Pr | Cl→Re | Pr→Ar | Pr→Cl | Pr→Re | Re→Ar | Re→Cl | Re→Pr | AVG |
|---|---|---|---|---|---|---|---|---|---|---|---|---|---|
| SourceRisk | 50.78 | 69.72 | 76.06 | 59.66 | 70.29 | 69.86 | 60.90 | 46.07 | **77.71** | 70.05 | **57.16** | 80.96 | 65.77 |
| IWCV | 50.24 | 69.72 | 77.28 | 62.63 | 67.24 | 69.86 | 58.84 | 49.69 | 75.72 | **71.45** | **57.16** | 79.97 | 65.82 |
| DEV | 51.07 | 69.72 | 76.64 | 59.66 | 67.24 | 71.26 | 58.84 | 49.69 | 75.72 | 70.95 | 50.65 | 76.64 | 64.84 |
| RV | 51.07 | 71.41 | 76.64 | 62.63 | 68.44 | 70.44 | 58.84 | 44.49 | **77.71** | **71.45** | 54.82 | **81.46** | 65.78 |
| Entropy | 45.93 | 69.72 | 75.49 | 55.29 | 67.22 | 68.35 | 54.26 | 43.30 | 75.69 | 70.00 | 49.99 | 80.60 | 62.99 |
| InfoMax | 50.47 | 69.72 | 75.49 | 62.46 | **70.98** | 68.35 | 61.23 | 43.30 | 75.69 | 70.00 | 55.37 | 80.60 | 65.31 |
| SND | 45.93 | 64.36 | 70.60 | 55.29 | 60.13 | 62.50 | 54.26 | 43.30 | 71.43 | 64.15 | 49.99 | 76.64 | 59.88 |
| Corr-C | 45.93 | 69.72 | 70.60 | 55.29 | 60.13 | 62.50 | 61.23 | 43.30 | 71.43 | **71.45** | 49.99 | 76.64 | 61.52 |
| EnsV-W | **51.73** | 72.07 | 76.64 | **64.65** | **70.98** | 71.26 | **63.66** | **50.52** | 77.48 | 70.99 | **57.16** | **81.46** | **67.38** |
| EnsV | 51.07 | **72.27** | **77.30** | 63.58 | 70.29 | **71.70** | 62.71 | 49.69 | **77.71** | **71.45** | 55.78 | 80.96 | 67.04 |
| Worst | 45.93 | 64.36 | 70.60 | 55.29 | 60.13 | 62.50 | 54.26 | 43.30 | 71.43 | 64.15 | 49.99 | 76.64 | 59.88 |
| Best | 51.73 | 72.27 | 77.30 | 64.65 | 70.98 | 71.70 | 63.66 | 50.52 | 77.71 | 71.45 | 57.16 | 81.46 | 67.55 |

Table 18: Accuracy (%) of closed-set UDA methods on *Office-31*.

| Method | ATDOC (Liang et al., 2021) | | | | | BNM (Cui et al., 2020) | | | | | CDAN (Long et al., 2018) | | | | |
|---|---|---|---|---|---|---|---|---|---|---|---|---|---|---|---|
| | A→D | A→W | D→A | W→A | AVG | A→D | A→W | D→A | W→A | AVG | A→D | A→W | D→A | W→A | AVG |
| SourceRisk | 88.96 | **87.80** | 73.65 | 71.46 | 80.47 | **90.36** | **89.43** | 73.13 | 72.70 | 81.41 | 91.16 | **89.06** | 66.33 | 61.46 | 77.00 |
| IWCV | 86.14 | 86.54 | 73.65 | 71.46 | 79.45 | 85.54 | **89.43** | 73.13 | 72.70 | 80.20 | 69.08 | 58.74 | 66.33 | 61.46 | 63.90 |
| DEV | 86.14 | 86.54 | 73.65 | 71.46 | 79.45 | 85.54 | **89.43** | 73.13 | 72.70 | 80.20 | 91.16 | 88.30 | 66.33 | 61.46 | 76.81 |
| RV | 89.96 | 87.23 | 74.28 | **75.58** | 81.76 | 88.55 | **89.43** | 74.90 | 66.52 | 79.85 | 91.16 | 88.30 | 76.18 | 70.36 | 81.50 |
| Entropy | 86.14 | **87.80** | 73.87 | 72.70 | 80.13 | 85.54 | 83.14 | 71.07 | 74.26 | 78.50 | 91.16 | **89.06** | 72.88 | **70.36** | 80.87 |
| InfoMax | 86.14 | **87.80** | 73.87 | 72.70 | 80.13 | 85.54 | 83.14 | 71.07 | 69.97 | 77.43 | 91.16 | 88.30 | 72.88 | **70.36** | 80.68 |
| SND | 92.37 | **87.80** | 73.87 | 72.70 | 81.69 | 85.54 | 83.14 | 74.62 | 74.26 | 79.39 | 92.37 | 88.55 | 72.88 | 70.22 | 81.01 |
| Corr-C | 90.96 | 84.40 | 71.88 | 70.22 | 79.37 | 84.34 | 78.99 | 67.80 | 66.52 | 74.41 | 67.67 | 59.62 | 58.15 | 58.43 | 60.97 |
| EnsV-W | 92.37 | **87.80** | 74.65 | 75.01 | **82.46** | 88.55 | **89.43** | 75.43 | **75.29** | 82.18 | **92.77** | 88.55 | 76.18 | 70.22 | **81.93** |
| EnsV | 90.96 | **87.80** | 74.65 | 75.01 | 82.11 | **90.36** | **89.43** | 75.43 | 74.30 | **82.38** | **92.77** | 88.55 | 76.18 | 70.22 | **81.93** |
| Worst | 86.14 | 84.40 | 71.88 | 70.22 | 78.16 | 84.34 | 78.99 | 67.80 | 66.52 | 74.41 | 67.67 | 57.11 | 58.15 | 58.43 | 60.34 |
| Best | 92.37 | 87.80 | 75.04 | 75.58 | 82.70 | 90.36 | 89.43 | 75.75 | 75.29 | 82.71 | 92.77 | 89.06 | 76.18 | 70.57 | 82.15 |

Table 19: Accuracy (%) of closed-set UDA methods on *Office-31*.

| Method | MCC (Jin et al., 2020) | | | | | MDD (Zhang et al., 2019) | | | | | SAFN (Xu et al., 2019) | | | | |
|---|---|---|---|---|---|---|---|---|---|---|---|---|---|---|---|
| | A→D | A→W | D→A | W→A | AVG | A→D | A→W | D→A | W→A | AVG | A→D | A→W | D→A | W→A | AVG |
| SourceRisk | 90.96 | 91.07 | 73.33 | 72.89 | 82.06 | 91.06 | 86.23 | 76.68 | 74.76 | 82.18 | 83.73 | 87.17 | 68.96 | 69.44 | 77.33 |
| IWCV | 91.16 | 88.55 | 73.33 | 72.89 | 81.48 | 91.16 | 89.18 | 76.68 | 74.30 | 82.83 | 86.55 | 80.38 | 68.96 | **69.68** | 76.39 |
| DEV | 89.16 | 93.08 | 73.33 | 72.06 | 81.91 | 91.16 | 89.18 | 76.68 | 74.62 | 82.91 | 86.55 | 80.38 | 68.96 | 67.45 | 75.84 |
| RV | 89.06 | 93.08 | 74.42 | 73.52 | 82.52 | **92.57** | 86.79 | 73.91 | **74.97** | 82.07 | 90.83 | 87.17 | 68.76 | 68.62 | 78.85 |
| Entropy | 90.56 | **93.46** | **74.83** | 73.02 | 82.97 | **92.57** | 90.82 | 78.03 | 74.58 | 84.00 | **91.57** | 85.66 | 67.20 | 69.26 | 78.42 |
| InfoMax | 89.16 | 88.55 | 74.16 | **73.70** | 81.39 | **92.57** | 90.82 | 78.03 | **74.97** | **84.10** | **91.57** | **87.42** | 67.20 | 69.26 | 78.86 |
| SND | **91.97** | **93.46** | **74.83** | 73.02 | **83.32** | 92.17 | 90.82 | 78.03 | **74.97** | 84.00 | 89.96 | 85.66 | 67.20 | 69.26 | 78.02 |
| Corr-C | 91.37 | **93.46** | **74.83** | 73.02 | 83.17 | 91.57 | 85.66 | 73.91 | 74.58 | 81.43 | 86.75 | 80.38 | 67.09 | **69.68** | 75.98 |
| EnsV-W | 90.56 | 91.07 | 74.16 | **73.70** | 82.37 | **92.57** | 90.82 | 77.53 | 74.30 | 83.80 | **91.57** | 87.17 | **70.22** | 69.12 | **79.52** |
| EnsV | 90.56 | 91.45 | 73.80 | **73.70** | 82.38 | **92.57** | 90.82 | 77.53 | 74.30 | 83.80 | 90.96 | 87.17 | **70.22** | 69.12 | 79.37 |
| Worst | 86.75 | 87.17 | 71.18 | 69.93 | 78.76 | 87.35 | 85.66 | 73.91 | 72.20 | 79.78 | 83.73 | 80.38 | 67.09 | 67.45 | 74.66 |
| Best | 91.97 | 93.46 | 74.83 | 74.01 | 83.57 | 92.57 | 92.20 | 78.03 | 75.01 | 84.45 | 91.57 | 87.42 | 70.43 | 69.68 | 79.78 |

Table 20: Accuracy (%) of a closed-set UDA method CDAN (Long et al., 2018) on *DomainNet-126*.

| Method | C → S | P → C | P → R | R → C | R → P | R → S | S → P | AVG |
|---|---|---|---|---|---|---|---|---|
| Entropy | **58.04** | **64.78** | 74.42 | **69.39** | **68.65** | **60.63** | **62.94** | **65.55** |
| InfoMax | **58.04** | **64.78** | 74.42 | **69.39** | **68.65** | **60.63** | **62.94** | **65.55** |
| SND | **58.04** | **64.78** | 74.42 | **69.39** | **68.65** | **60.63** | 60.70 | 65.23 |
| Corr-C | **58.04** | 57.73 | 74.42 | 56.98 | 65.07 | 51.23 | 60.70 | 60.60 |
| EnsV-W | 55.15 | 60.98 | 73.86 | 60.99 | 65.07 | 55.50 | 60.27 | 61.69 |
| EnsV | 56.73 | 64.67 | **74.44** | 67.08 | 67.97 | 58.12 | 62.57 | 64.51 |
| Worst | 51.59 | 57.73 | 73.44 | 56.98 | 63.06 | 51.23 | 58.46 | 58.93 |
| Best | 58.04 | 64.78 | 74.44 | 69.39 | 68.65 | 60.63 | 62.94 | 65.55 |

Table 21: Accuracy (%) of a closed-set UDA method BNM (Cui et al., 2020) on *DomainNet-126*.

| Method | C → S | P → C | P → R | R → C | R → P | R → S | S → P | AVG |
|---|---|---|---|---|---|---|---|---|
| Entropy | 56.42 | 61.57 | 74.31 | 65.15 | 65.15 | 40.95 | 63.42 | 61.00 |
| InfoMax | 56.42 | 68.95 | 74.31 | 65.15 | 65.15 | 54.93 | 63.42 | 64.05 |
| SND | 43.78 | 61.57 | 74.31 | 51.55 | 54.40 | 40.95 | 54.59 | 54.45 |
| Corr-C | 43.78 | 60.03 | **77.62** | 59.47 | 67.19 | 40.95 | 59.64 | 58.38 |
| EnsV-W | **58.48** | 68.42 | **77.62** | 66.05 | **67.79** | 57.65 | **64.34** | 65.76 |
| EnsV | 57.73 | **69.63** | **77.62** | **66.10** | **67.79** | 57.65 | **64.34** | **65.84** |
| Worst | 43.78 | 60.03 | 74.31 | 51.55 | 54.40 | 40.95 | 54.59 | 54.23 |
| Best | 58.48 | 69.63 | 78.68 | 66.10 | 67.79 | 58.50 | 65.20 | 66.34 |

Table 22: Accuracy (%) of a closed-set UDA method ATDOC (Liang et al., 2021) on *DomainNet-126*.

| Method | C → S | P → C | P → R | R → C | R → P | R → S | S → P | AVG |
|---|---|---|---|---|---|---|---|---|
| Entropy | 46.43 | 65.98 | 79.60 | 61.52 | 64.24 | 57.92 | 59.46 | 62.16 |
| InfoMax | 46.43 | 65.98 | 79.60 | 61.52 | 64.24 | 57.92 | 59.46 | 62.16 |
| SND | 46.43 | 65.98 | 79.60 | 61.52 | 64.24 | 47.58 | 59.46 | 60.69 |
| Corr-C | 54.71 | 60.63 | 74.42 | 59.33 | 64.58 | 52.66 | 59.95 | 60.90 |
| EnsV-W | **63.12** | 69.57 | 78.33 | 67.93 | 69.32 | 60.85 | 66.33 | 67.92 |
| EnsV | 62.11 | **71.14** | **80.01** | **69.45** | **69.79** | 61.35 | **67.10** | **68.71** |
| Worst | 46.43 | 60.63 | 74.42 | 59.33 | 64.24 | 47.58 | 59.46 | 58.87 |
| Best | 63.12 | 71.14 | 80.38 | 69.45 | 69.79 | 61.35 | 67.10 | 68.90 |

Table 23: Accuracy (%) of a partial-set UDA method PADA (Cao et al., 2018) on *Office-Home*.

| Method | Ar → Cl | Ar → Pr | Ar → Re | Cl → Ar | Cl → Pr | Cl → Re | Pr → Ar | Pr → Cl | Pr → Re | Re → Ar | Re → Cl | Re → Pr | AVG |
|---|---|---|---|---|---|---|---|---|---|---|---|---|---|
| SourceRisk | 45.03 | 68.85 | 81.89 | 43.25 | 46.83 | 57.26 | 57.21 | 36.42 | 76.53 | 71.26 | 44.30 | 77.76 | 58.87 |
| IWCV | **55.58** | 65.10 | 84.54 | 51.42 | **61.29** | 53.01 | 56.93 | 35.16 | 81.34 | 70.52 | **60.78** | 74.12 | 62.49 |
| DEV | 54.81 | **78.15** | 78.02 | **58.13** | **61.29** | 50.14 | 67.86 | 35.16 | 83.21 | 74.66 | 57.91 | 77.76 | 64.76 |
| RV | 43.22 | 65.10 | 81.89 | 42.70 | 48.74 | 52.79 | 57.21 | 35.16 | 77.80 | 73.46 | 44.30 | 77.76 | 58.34 |
| Entropy | 40.12 | 40.11 | 55.94 | 52.43 | 37.25 | 50.14 | 57.30 | 47.22 | 81.34 | 70.52 | 52.18 | 82.13 | 55.56 |
| InfoMax | 54.81 | 69.24 | 78.02 | 52.43 | 37.25 | 50.14 | 57.30 | 47.22 | 71.84 | 70.52 | 52.18 | 74.12 | 59.59 |
| SND | 40.12 | 40.11 | 55.94 | 58.13 | 56.13 | 64.11 | 70.62 | **51.22** | 81.34 | 74.66 | **60.78** | 82.13 | 61.27 |
| Corr-C | 40.12 | 40.11 | 55.94 | 54.18 | 46.89 | 53.01 | 58.59 | 38.93 | 77.80 | 71.26 | 57.91 | 77.70 | 56.04 |
| EnsV-W | **55.58** | 77.25 | 86.14 | **58.13** | 60.17 | **67.86** | **73.00** | 37.97 | **84.04** | 76.77 | 57.91 | 83.75 | **68.21** |
| EnsV | 54.81 | 69.24 | **86.53** | **58.13** | 56.13 | 64.11 | 70.62 | **51.22** | 84.04 | **76.86** | **60.78** | **84.20** | 68.06 |
| Worst | 40.12 | 40.11 | 55.94 | 41.41 | 37.25 | 50.14 | 56.93 | 34.87 | 71.84 | 70.52 | 44.24 | 74.12 | 51.46 |
| Best | 55.58 | 78.15 | 86.53 | 58.13 | 61.29 | 68.19 | 73.00 | 51.22 | 84.04 | 76.86 | 60.78 | 84.20 | 69.83 |

Table 24: Accuracy (%) of a partial-set UDA method SAFN (Xu et al., 2019) on *Office-Home*.

| Method | Ar → Cl | Ar → Pr | Ar → Re | Cl → Ar | Cl → Pr | Cl → Re | Pr → Ar | Pr → Cl | Pr → Re | Re → Ar | Re → Cl | Re → Pr | AVG |
|---|---|---|---|---|---|---|---|---|---|---|---|---|---|
| SourceRisk | **59.40** | 77.14 | 81.34 | 63.97 | 67.00 | 71.29 | 65.60 | 46.21 | 76.81 | 70.89 | 58.51 | 79.10 | 68.11 |
| IWCV | 52.24 | 74.45 | **82.16** | 70.98 | 62.41 | 70.18 | 63.45 | 53.49 | 76.81 | 73.65 | 56.00 | 78.49 | 67.86 |
| DEV | 55.22 | 74.45 | 80.07 | 70.98 | 67.00 | 71.29 | 65.70 | 51.70 | 76.81 | 73.65 | 57.91 | 80.39 | 68.58 |
| RV | 53.67 | 71.60 | 81.34 | 67.58 | 67.00 | 73.27 | 65.70 | 48.54 | 76.81 | 73.65 | 56.00 | 79.89 | 67.92 |
| Entropy | 58.93 | 74.90 | 80.73 | 70.98 | **74.12** | 69.80 | 70.16 | 50.09 | 79.24 | 74.10 | 57.85 | 80.06 | 70.08 |
| InfoMax | 51.82 | 67.62 | 76.97 | 64.65 | 65.77 | 69.80 | 59.69 | 50.09 | 74.10 | 66.67 | 53.31 | 75.52 | 64.67 |
| SND | 51.82 | 74.90 | 80.73 | 70.98 | **74.12** | **75.10** | 70.16 | 50.09 | 79.24 | 74.10 | 53.31 | 80.06 | 69.55 |
| Corr-C | **59.40** | **77.20** | **82.16** | 72.89 | 72.89 | **75.10** | 70.16 | **55.70** | 80.12 | **75.94** | 52.00 | **80.73** | 70.75 |
| EnsV-W | **59.40** | **77.20** | **82.16** | 71.72 | 72.89 | 74.82 | **72.45** | **55.70** | **80.73** | **75.94** | **59.16** | **80.73** | **71.91** |
| EnsV | 55.22 | 76.30 | 81.28 | 67.58 | 70.31 | 74.05 | 70.16 | 54.63 | 80.12 | 75.21 | 58.51 | 80.39 | 70.31 |
| Worst | 51.52 | 67.62 | 76.97 | 61.07 | 62.35 | 69.80 | 59.69 | 46.21 | 74.10 | 66.67 | 52.00 | 75.52 | 63.63 |
| Best | 59.40 | 77.20 | 82.16 | 71.72 | 74.12 | 75.10 | 72.45 | 55.70 | 80.73 | 75.94 | 59.16 | 80.73 | 72.03 |

Table 25: H-score (Fu et al., 2020; Bucci et al., 2020) (%) of an open-partial-set UDA method DANCE (Saito et al., 2020) on *Office-Home*.

| Method | Ar → Cl | Ar → Pr | Ar → Re | Cl → Ar | Cl → Pr | Cl → Re | Pr → Ar | Pr → Cl | Pr → Re | Re → Ar | Re → Cl | Re → Pr | AVG |
|---|---|---|---|---|---|---|---|---|---|---|---|---|---|
| Entropy | 38.29 | 26.08 | 36.51 | 32.92 | 17.10 | 32.19 | 37.69 | 46.40 | 45.53 | 25.39 | 33.75 | 39.37 | 34.27 |
| InfoMax | 38.29 | 26.08 | 36.51 | 32.92 | 17.10 | 32.19 | 37.69 | 46.40 | 45.33 | 25.39 | 33.75 | 39.37 | 34.25 |
| SND | 1.00 | 0.00 | 12.73 | 0.00 | 42.84 | 1.95 | 19.77 | 11.99 | 35.69 | 25.39 | 0.00 | 28.40 | 14.98 |
| Corr-C | 1.00 | 0.00 | 12.73 | 0.00 | 42.84 | 1.95 | 19.77 | 11.99 | 35.69 | 69.02 | 0.00 | 28.40 | 18.62 |
| EnsV-W | **67.00** | 75.15 | **66.57** | 67.87 | 67.35 | 59.05 | 66.41 | **62.59** | **69.40** | 59.86 | **67.54** | 73.40 | 66.85 |
| EnsV | 38.40 | **76.96** | **66.57** | **71.76** | **75.17** | 69.99 | **77.42** | 48.15 | **69.40** | **81.84** | **67.54** | **84.31** | **68.96** |
| Worst | 1.00 | 0.00 | 12.73 | 0.00 | 17.10 | 1.95 | 19.77 | 11.99 | 35.69 | 25.39 | 0.00 | 28.40 | 12.84 |
| Best | 67.00 | 76.96 | 66.57 | 71.76 | 75.17 | 69.99 | 77.42 | 64.32 | 72.87 | 81.84 | 67.54 | 84.31 | 72.98 |

Table 26: Accuracy (%) of a source-free UDA method SHOT (Liang et al., 2020) on *Office-31*.

| Method | A → D | A → W | D → A | W → A | AVG |
|---|---|---|---|---|---|
| Entropy | 90.76 | 88.68 | 71.21 | 72.13 | 80.69 |
| InfoMax | 90.76 | 88.68 | 71.21 | 72.13 | 80.69 |
| SND | 90.76 | 88.68 | 71.21 | 72.13 | 80.69 |
| Corr-C | 90.76 | 90.19 | 71.21 | 71.96 | 81.03 |
| EnsV-W | **94.78** | **91.82** | **75.15** | **74.55** | **84.08** |
| EnsV | **94.78** | **91.82** | **75.15** | **74.55** | **84.08** |
| Worst | 90.76 | 88.68 | 71.21 | 71.92 | 80.64 |
| Best | 94.78 | 93.33 | 75.58 | 74.55 | 84.56 |

Table 27: Worst-case selections for various target-only model selection methods, reported as the H-score (%) for OPDA and accuracy (%) for other tasks, demonstrate that EnsV consistently avoids the worst selections, while other methods often encounter significant challenges.

| Method | CDA | | | | | | PDA | | OPDA | | SFUDA | |
|---|---|---|---|---|---|---|---|---|---|---|---|---|
| | ATDOC | ATDOC | BNM | BNM | MDD | SAFN | PADA | PADA | DANCE | DANCE | SHOT | DINE |
| | Cl→Ar | C→S | Ar→Pr | R→S | Pr→Cl | Pr→Cl | Ar→Re | Re→Ar | Re→Ar | Pr→Re | D→A | T→V |
| Entropy | 59.25 | 46.43 | 67.04 | 40.95 | 55.85 | 43.30 | 55.94 | 70.52 | 25.39 | 45.53 | 71.21 | 71.99 |
| InfoMax | 59.25 | 46.43 | 67.04 | 54.93 | 55.85 | 43.30 | 78.02 | 70.52 | 25.39 | 45.53 | 71.21 | 71.99 |
| SND | 59.25 | 46.43 | 67.04 | 40.95 | 21.60 | 43.30 | 55.94 | 74.66 | 25.39 | 35.69 | 71.21 | 74.43 |
| Corr-C | 59.37 | 54.71 | 67.06 | 40.95 | 21.60 | 43.30 | 55.94 | 71.26 | 69.02 | 35.69 | 71.21 | 71.99 |
| EnsV | **66.25** | **62.11** | **77.00** | **57.65** | **57.02** | **49.69** | **86.53** | **76.86** | **81.84** | **69.40** | **75.15** | **74.43** |
| Worst | 59.25 | 46.43 | 67.04 | 40.95 | 21.60 | 43.30 | 55.94 | 70.52 | 25.39 | 35.69 | 71.21 | 71.99 |
| Best | 66.91 | 63.12 | 77.00 | 58.50 | 57.02 | 50.52 | 86.53 | 76.86 | 81.84 | 72.87 | 75.15 | 76.17 |

