# OpenReview forum: "Simplifying and Stabilizing Model Selection in Unsupervised Domain Adaptation"
_ICLR.cc/2024/Conference — ICLR 2024 Conference Withdrawn Submission_

### Official Review · Reviewer_x2YN · 2023-10-19

**Soundness:** 2 fair
**Presentation:** 2 fair
**Contribution:** 2 fair
**Rating:** 3
**Confidence:** 4

**Summary:**

The paper proposes a simple yet effective method for model selection under unsupervised domain adaptation. The authors introduce an ensemble-based method to better approximate the true labels on target domain to utilize model parameter selection. Comprehensive empirical studies are done over multiple UDA methods and datasets.

**Strengths:**

1. The paper is the first work to do model selection under UDA via ensembling. Ensembling is used for better approximation of target labels.
2. Sufficient experiments were done to justify the effectiveness of the proposed method.

**Weaknesses:**

I have the following concerns that lowers my ratings for this work:
1. The method seems quite trivial for me, as it trade memory and time for performance. Suppose we have $n$ set of hyper-parameters to select from, using this method still requires to perform $n$ training processes on source and target data, saving at least $n$ models, and $n$ inference processes on target data.
2. Though effective in practice, the paper lacks proper analysis of why it works so well. I list one of my questions here: In $n$ models, if only one of them works well but the others do not, won't the $n-1$ models' predictions dominate the final prediction and the correct model be not selected?
3. Overclaims: In terms of the analysis and results, some claimed facts are not true. For example, in section 4.2 PDA: "Our EnsV outperforms all other model selection methods by a significant margin in terms of average accuracy", which is not true given there results in Table 5 SAFN.
4. While the authors mentioned EnsV is a "stable" method, I do not see a justification for stability.
4. Is Figure 1 partly drawn by hand?
5. Inconsistent result presentation. Table 4,8 misses some baseline comparisons, in contrast to other tables.

**Questions:**

See weaknesses.

**Details Of Ethics Concerns:**

None.

---

### Official Review · Reviewer_y3mw · 2023-10-26

**Soundness:** 2 fair
**Presentation:** 3 good
**Contribution:** 3 good
**Rating:** 5
**Confidence:** 4

**Summary:**

This work proposes a model selection method for UDA. From a model candidate pool, the authors first use the prediction ensemble as a role model, which serves as an estimate of the oracle target performance, then choose the model that performs the most similarly to the role model. The proposed method consistently outperforms other model selection methods for UDA in various settings.

**Strengths:**

1. The authors perform extensive experiments to validate the empirical performance of the proposed algorithm. The algorithm indeed performs consistently compared with other model selection methods across various domain adaptation settings.

2. The algorithm is simple and does not require significant computation overhead.

3. The model selection problem is important and under-explored in the UDA community.

**Weaknesses:**

1. There is no reason to believe that the ensemble model is a reliable estimate of the target ground-truth. The “theoretical analysis” in section 3.1 does not justify the point because “no worse than the worst model” can be achieved by any algorithm.

2. The authors do not provide adequate analysis of the failure case of the method. The ensemble approach highly relies on the property of the candidate pool, and the authors should discuss this point in more detail about the success criteria of the proposed method.

**Questions:**

1. I do not understand the purpose of the “theoretical analysis” part. Since the model will be selected from the candidate pool anyway, why do the authors prove that the role model cannot be worse than the worst candidate? The worst-case scenario is just to select the worst candidate, right? What is the message to convey here?

2. For scenarios where unseen classes can be presented in the target domain, why is directly applying the ensemble still reliable? How is the ensemble score $f(\theta,x)$ calculated?

---

### Official Review · Reviewer_DSsh · 2023-11-02

**Soundness:** 2 fair
**Presentation:** 3 good
**Contribution:** 2 fair
**Rating:** 5
**Confidence:** 4

**Summary:**

This paper addresses the model selection problem in UDA by introducing an ensemble-based method, EnsV. This approach guarantees performance that surpasses the worst candidate model. Experimental evaluations further establish EnsV's superior performance over existing methods.

**Strengths:**

1.	The approach is straightforward to implement.
2.	EnsV is able to deal with covariate shift and label shift and does not require extra training.
3.	The paper is clearly written.

**Weaknesses:**

1.	Although EnsV ensures it won't select the worst-performing model, it doesn't guarantee optimal performance. Given the significance of performance in UDA, this limitation is noteworthy.
2.	Proposition 1 is grounded on the assumption that NLL is the loss function. However, in UDA, a common loss function is the upper bound for the target classification loss, as outlined by Ben-David, Shai [1]. Does Proposition 1 still apply when not using NLL as the loss function?

[1] Ben-David, Shai, et al. "A theory of learning from different domains." Machine learning 79 (2010): 151-175.

**Questions:**

1.	In your model selection on various UDA methods, have you considered performing model selection on more recent methods with SOTA performance on Office-Home, such as PMtrans, MIC, ELS, SDAT, and CDTrans mentioned in [2]?

[2] https://paperswithcode.com/sota/domain-adaptation-on-office-home

2.	It would be beneficial to compare EnsV with some hyperparameter optimization tools, like the Ax Platform [3]. As the paper outlined in the Method section, the model selection problem in UDA is essentially equivalent to the hyperparameter selection challenge.

[3] https://ax.dev/docs/why-ax.html

3.	Rather than using an equal weight to ensemble models, have you considered assigning weights to potentially improve results?

4.	In the section of “robustness to bad candidates”, if you randomly sample 5 checkpoints and ensemble these, how does the accuracy fluctuate?